# Deep Gaussian Markov Random Fields for Graph-Structured Dynamical Systems

**Fiona Lippert**
University of Amsterdam
f.lippert@uva.nl

**Bart Kranstauber**
University of Amsterdam
b.kranstauber@uva.nl

**E. Emiel van Loon**
University of Amsterdam
e.e.vanloon@uva.nl

**Patrick Forré**
University of Amsterdam
p.d.forre@uva.nl

## Abstract

Probabilistic inference in high-dimensional state-space models is computationally challenging. For many spatiotemporal systems, however, prior knowledge about the dependency structure of state variables is available. We leverage this structure to develop a computationally efficient approach to state estimation and learning in graph-structured state-space models with (partially) unknown dynamics and limited historical data. Building on recent methods that combine ideas from deep learning with principled inference in Gaussian Markov random fields (GMRF), we reformulate graph-structured state-space models as Deep GMRFs defined by simple spatial and temporal graph layers. This results in a flexible spatiotemporal prior that can be learned efficiently from a single time sequence via variational inference. Under linear Gaussian assumptions, we retain a closed-form posterior, which can be sampled efficiently using the conjugate gradient method, scaling favourably compared to classical Kalman filter based approaches.

## 1 Introduction

Consider the problem of monitoring air pollution, the leading environmental risk factor for mortality worldwide [11, 42]. In the past years, sensor networks have been installed across major metropolitan areas, measuring the concentration of pollutants across time and space [27, 38]. These measurements are, however, prone to noise or might be missing completely due to hardware failures or limited sensor coverage. To identify sources of pollution or travel routes with limited exposure, it is essential to recover pollution levels and provide principled uncertainty estimates for unobserved times and locations. Similar problems occur also in the geosciences, ecology, neuroscience, epidemiology, or transportation systems, where animal movements, the spread of diseases, or traffic load need to be estimated from imperfect data to facilitate scientific discovery and decision-making. From a probabilistic inference perspective, all these problems amount to estimating the posterior distribution over the latent states of a spatiotemporal system given partial and noisy multivariate time series data.

To make inference in these systems feasible, it is common to assume a state-space model, where the latent states evolve according to a Markov chain. Then, the classical Kalman filter (KF) [29] or its variants [47, 37] can be used for efficient inference, scaling linearly with the time series length. However, the complexity with respect to the state dimension remains cubic due to matrix-matrix multiplications and inversions. Thus, as the spatial coverage of available measurements and thereby the dimensionality of the state variables increase, KF-based approaches quickly become computationally prohibitive. This is especially problematic in cases where the underlying dynamics are (partially) unknown and need to be learned from data, requiring repeated inference during the

37th Conference on Neural Information Processing Systems (NeurIPS 2023).

optimization loop [21, 40, 9]. If, additionally, the access to historical training data is limited, it is essential to incorporate prior knowledge for learning and inference to remain both data-efficient and computationally feasible [56].

For spatiotemporal systems, prior knowledge is often available in the form of graphs, representing the dependency structure of state variables. For air pollution, this could be the city road network and knowledge about wind directions which influence the spread of particulate matter. Incorporating this knowledge into the initial state distribution and the transition model of a state-space model results in a Dynamic Bayesian network (DBN) [12, 39, 56], an interpretable and data-efficient graphical model for multivariate time series data. If the joint space-time graph is sparse and acyclic, belief propagation methods allow for fast and principled inference of marginal posteriors in DBNs, scaling linear with the number of edges [45, 39]. However, when the underlying dynamics are complex, requiring denser graph structures and flexible error terms with a loopy spatial structure, the scalability and convergence of these methods is no longer guaranteed.

Combining the favorable statistical properties of classical graphical models with the flexibility and scalability of deep learning offers promising solutions in this regard [57]. Deep Gaussian Markov random fields (DGMRF), for example, integrate principled inference in Gaussian Markov random fields (GMRF) with convolutional [50] or graph layers [43] to define a new flexible family of GMRFs that can be learned efficiently by maximizing a variational lower bound, while facilitating principled Bayesian inference that remains computationally feasible even for high-dimensional complex models with dense dependency structures. While in principle DGMRFs can be applied to spatiotemporal systems by treating each time step independently, they are by design limited to spatial structures.

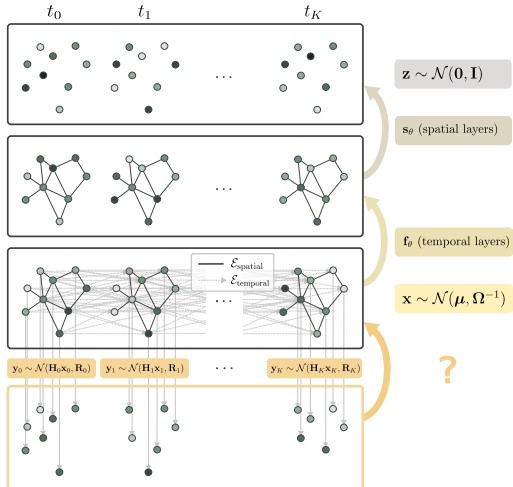

Figure 1: ST-DGMRF overview. We reconstruct the latent states of a graph-structured dynamical system from partial and noisy observations (orange arrow with question mark). Temporal and spatial layers transform the state $\mathbf{x}$ to a standard Gaussian, which implicitly defines a space-time GMRF prior.

In this paper we extend the DGMRF framework to spatiotemporal systems with graph-structured states, transitions and noise terms. To this end, we formulate a dynamical prior based on a DBN that relates state variables over adjacent time steps, and DGMRFs that capture the spatial structure of unmodeled exogenous influences and errors in the transition model. The key to our approach is the insight that, using locally linear Gaussian transition models, the precision matrix of the joint space-time process preserves the structure of transition and noise precision matrices. This is in contrast to the marginal distributions used in Kalman filter based approaches for which both precision and covariance matrices become dense over time. A convenient factorization of the joint precision matrix then leads to a spatiotemporal DGMRF formulation that is equivalent to the generative state-space formulation, but inherits the favorable properties of DMGRFs for learning and inference.

## 1.1 Related work

The simplest approach to scaling Kalman filtering and smoothing to high-dimensional state variables is to assume a decomposition into subprocesses, that can be reconstructed independently [30]. For spatiotemporal processes, however, this assumption can be detrimental. Sample-based approaches, like the Ensemble KF [17, 18, 26, 31], instead approximate state distributions with Monte Carlo estimates, balancing computational complexity and accuracy. This is widely used in the geosciences [53, 55] where accurate models of the underlying process are available. If the dynamics are (partially) unknown, the parameters are typically estimated jointly with the system states via state augmentation [2]. However, this becomes problematic if the transition model is only a poor approximation and complex error covariances need to be estimated [15, 52]. Alternatively, expectation maximization

(EM) is used to estimate parameters iteratively [21], which requires repeated ensemble simulations and thus quickly becomes computationally demanding.

In machine learning, neural network-based approaches map high-dimensional data to a latent space that is either low-dimensional [19] or factorizes conveniently [7], such that standard KF inference remains feasible. To learn suitable maps, however, these methods require sufficiently large training data sets. Moreover, the projection to an abstract latent space hinders the incorporation of structural or functional prior knowledge. Alternatively, variational inference allows for fast inference in (deep) state space models [20, 3, 28, 34, 54] by approximating the true posterior with a computationally convenient variational distribution. While this approach is highly flexible and scalable, it is known to severely underestimate uncertainties [5], which can be detrimental for science and decision-making.

Gaussian processes (GP) are another widely used model class for spatiotemporal inference [35]. In contrast to KF-based approaches, GPs encode prior knowledge into their covariance kernel. To increase the expressivity beyond standard kernels and overcome computational limitations, GPs have been combined with deep learning [14, 13]. However, scaling GPs to large-scale spatiotemporal processes with graph structure remains challenging. Scalable approaches for temporal data involve reformulating GPs as state-space models and applying KF-based recursions [24, 49, 10, 1, 58, 23] which, as discussed before, remains computationally prohibitive for large state spaces. Extensions to graph-structured domains [8, 41] rely on the relation between GPs and stochastic differential equations (SDE) [36, 51], which makes it cumbersome to design new kernels or to incorporate prior knowledge that cannot be formulated as an SDE.

## 1.2 Our contributions

In contrast to most previous approaches, we focus on settings where the dynamics are (partially) unknown and need to be learned from data, while historical training data is limited or even unavailable. Our contributions can be summarized as follows:

1. We propose ST-DGMRF, which extends the DGMRF framework to spatiotemporal systems, by reformulating graph-structured state-space models as joint space-time GMRFs that are implicitly defined by simple spatial and temporal graph layers.

2. We show that the multi-layer space-time formulation facilitates efficient learning and inference, scaling linearly w.r.t. the number of layers, time series length, and state dimensionality.

3. In experiments on synthetic and real world data, we demonstrate that our approach provides accurate state and uncertainty estimates, comparing favorably to other scalable approaches relying on ensembles or simplifications of the dependency structure.

## 2 Preliminaries

### 2.1 Problem formulation

Consider a single sequence of high-dimensional measurements $\mathbf{y}_{0:K} = (\mathbf{y}_0, \ldots, \mathbf{y}_K)$ taken at consecutive time points $t_0, \ldots, t_K$. Each $\mathbf{y}_k \in \mathbb{R}^{M_k}$ represents partial and noisy observations (e.g. from spatially distributed sensors) generated according to a linear Gaussian noise model

$$\mathbf{y}_k = \mathbf{H}_k \mathbf{x}_k + \boldsymbol{\xi}_k, \quad \boldsymbol{\xi}_k \sim \mathcal{N}(\mathbf{0}, \mathbf{R}_k) \tag{1}$$

from the latent state $\mathbf{x}_k \in \mathbb{R}^N$ of an underlying dynamical process of dimensionality $N \geq M_k$ (e.g. air pollution along $N$ roads within an urban area). It is common to assume either $\mathbf{R}_k = \text{diag}(\sigma_{k,1}^2, \ldots, \sigma_{k,M_k}^2)$, representing independent sensors with varying noise levels, or $\mathbf{R}_k = \sigma^2 \mathbf{I}_{M_k}$, representing a fixed noise level across all sensors and time points [9]. Further, the observation matrices $\mathbf{H}_k \in \mathbb{R}^{M_k \times N}$ are typically very sparse with $\mathcal{O}(M_k)$ number of non-zero entries.

Assuming that all $\mathbf{H}_k$ are known, we aim at reconstructing the latent system states $\mathbf{x}_{0:K} = (\mathbf{x}_0, \ldots, \mathbf{x}_K) \in \mathbb{R}^{(K+1) \times N}$ from data $\mathbf{y}_{0:K}$. Since observations are partial and noisy, $\mathbf{x}_{0:K}$ cannot be recovered without ambiguity. It is therefore essential to treat the problem probabilistically and incorporate available domain knowledge into the prior distribution $p(\mathbf{x}_{0:K} \mid \boldsymbol{\theta})$. The task then amounts to (i) specifying a suitable prior that captures the spatiotemporal structure of $\mathbf{x}_{0:T}$, (ii) estimating unknown parameters, and (iii) computing the posterior distribution $p(\mathbf{x}_{0:K} \mid \mathbf{y}_{0:K}, \hat{\boldsymbol{\theta}})$.

**Notation** We use $\mathbf{x} = \text{vec}(\mathbf{x}_{0:K}) \in \mathbb{R}^{(K+1)N}$ and $\mathbf{y} = \text{vec}(\mathbf{y}_{0:K}) \in \mathbb{R}^M$ with $M = \sum_{k=0}^K M_k$ to denote vectorized states and measurements. Similarly, we use $\mathbf{H} = \text{diag}(\mathbf{H}_0, \ldots, \mathbf{H}_K) \in \mathbb{R}^{M \times (K+1)N}$ and $\mathbf{R} = \text{diag}(\mathbf{R}_0, \ldots, \mathbf{R}_K) \in \mathbb{R}^{M \times M}$ to denote the joint spatiotemporal observation and covariance matrix.

In the following sections, we briefly introduce two graphical models, linear dynamical systems and Gaussian Markov random fields, which form the temporal and spatial backbone of our method. We then introduce the deep GMRF framework on which we build in Section 3 to formulate a flexible multi-layer spatiotemporal prior.

## 2.2 Linear dynamical system

A discrete-time linear dynamical system (LDS) is a state-space model that defines the prior over states $\mathbf{x}_{0:K}$ in terms of linear Gaussian transition models

$$\mathbf{x}_k = \mathbf{F}_k \mathbf{x}_{k-1} + \mathbf{c}_k + \boldsymbol{\epsilon}_k, \quad \boldsymbol{\epsilon}_k \sim \mathcal{N}(\mathbf{0}, \mathbf{Q}_k^{-1}) \quad \forall k \in \{1, \ldots, K\} \tag{2}$$

and an initial distribution $\mathbf{x}_0 \sim \mathcal{N}(\boldsymbol{\mu}_0, \mathbf{Q}_0^{-1})$. The classical Kalman filter [29] and its variants [47, 37] use recursive algorithms that exploit the temporal independence structure of the prior and its conjugacy to the observation model (see Eq. (1)) to compute marginal posterior distributions $p(\mathbf{x}_k \mid \mathbf{y}_{0:k})$ in $\mathcal{O}(KN^3)$. The same formulas can be used within the EM-algorithm to learn unknown parameters $\boldsymbol{\mu}_0, \mathbf{Q}_0, \mathbf{F}_k, \mathbf{c}_k, \mathbf{Q}_k^{-1}$ in $\mathcal{O}(JKN^3)$ for $J$ iterations. While this scales favorably for long time series, it remains computationally prohibitive for high-dimensional systems with $N \gg K$.

## 2.3 Gaussian Markov random fields

A multivariate Gaussian $\mathcal{N}(\boldsymbol{\mu}, \boldsymbol{\Omega}^{-1})$ forms a GMRF [48] with respect to an undirected graph $\mathcal{G} = (\mathcal{V}, \mathcal{E})$ if the edges $\mathcal{E}$ correspond to the non-zero entries of the precision matrix $\boldsymbol{\Omega}$, i.e. $\boldsymbol{\Omega}_{ij} \neq 0 \Leftrightarrow (i, j) \in \mathcal{E}$. Given a linear Gaussian observation model $\mathbf{y} \sim \mathcal{N}(\mathbf{Hx}, \mathbf{R})$, the posterior can be written in closed form $p(\mathbf{x} \mid \mathbf{y}, \hat{\boldsymbol{\theta}}) = \mathcal{N}(\boldsymbol{\mu}^+, (\boldsymbol{\Omega}^+)^{-1})$ with precision matrix $\boldsymbol{\Omega}^+ = \boldsymbol{\Omega} + \mathbf{H}^T \mathbf{R}^{-1} \mathbf{H}$ and mean $\boldsymbol{\mu}^+ = (\boldsymbol{\Omega}^+)^{-1}(\boldsymbol{\Omega}\boldsymbol{\mu} + \mathbf{H}^T \mathbf{R}^{-1} \mathbf{y})$.

However, naively computing $\boldsymbol{\mu}^+$ by matrix inversion requires $\mathcal{O}(K^3 N^3)$ computations and $\mathcal{O}(K^2 N^2)$ memory, which is infeasible for high-dimensional systems. Instead, the conjugate gradient method [6] can be used to iteratively solve the sparse linear system $\boldsymbol{\Omega}^+ \boldsymbol{\mu}^+ = \boldsymbol{\Omega}\boldsymbol{\mu} + \mathbf{H}^T \mathbf{R}^{-1} \mathbf{y}$ for $\boldsymbol{\mu}^+$. The same approach allows to generate samples $\hat{\mathbf{x}} \sim \mathcal{N}(\boldsymbol{\mu}^+, (\boldsymbol{\Omega}^+)^{-1})$ and obtain Monte Carlo estimates of marginal variances [44]. In practice, however, it remains difficult to design suitable precision matrices that are expressive but sparse enough to remain computationally feasible.

## 2.4 Deep Gaussian Markov random fields

A DGMRF [50] is defined by an affine transformation

$$\mathbf{z} = \mathbf{g}_\theta(\mathbf{x}) = \mathbf{G}_\theta \mathbf{x} + \mathbf{b}_\theta \quad \text{with} \quad \mathbf{z} \sim \mathcal{N}(\mathbf{0}, \mathbf{I}), \tag{3}$$

where $\mathbf{g}_\theta = \mathbf{g}_\theta^{(L)} \circ \cdots \circ \mathbf{g}_\theta^{(1)}$ is a composition of $L$ simple linear layers with convenient computational properties. This implicitly defines a GMRF $\mathbf{x} \sim \mathcal{N}(\boldsymbol{\mu}, \boldsymbol{\Omega}^{-1})$ with $\boldsymbol{\mu} = -\mathbf{G}_\theta^{-1} \mathbf{b}_\theta$ and $\boldsymbol{\Omega} = \mathbf{G}_\theta^T \mathbf{G}_\theta$. The multi-layer construction facilitates fast and statistically sound posterior inference using the conjugate gradient method, as well as fast parameter learning using a variational approximation, even when the resulting precision matrix $\boldsymbol{\Omega}$ becomes dense.

While originally, [50] defined their layers only for lattice-structured data, [43] generalized this to arbitrary graphs. Central to their approach is that each layer $\mathbf{h}^{(l)} = \mathbf{G}_\theta^{(l)} \mathbf{h}^{(l-1)} + \mathbf{b}_\theta^{(l)}$ is defined on a sparse base graph $\bar{\mathcal{G}}$ with adjacency matrix $\mathbf{A}$ and degree matrix $\mathbf{D}$ such that

$$\mathbf{G}^{(l)} = \alpha_l \mathbf{D}^{\gamma_l} + \beta_l \mathbf{D}^{\gamma_l - 1} \mathbf{A} \quad \text{and} \quad \mathbf{b}_\theta^{(l)} = b_l \mathbf{1} \tag{4}$$

with parameters $\boldsymbol{\theta}_l = (\alpha_l, \beta_l, \gamma_l, b_l)$. This construction allows for fast log-determinant computations during learning, and GPU-accelerated computation of $\mathbf{g}_\theta(\mathbf{x})$ using existing software for graph neural networks. Note that for $L$ layers defined on $\bar{\mathcal{G}}$, the sparsity pattern of $\boldsymbol{\Omega}$ corresponds to the sparsity pattern of $(\mathbf{A} + \mathbf{D})^{2L}$. And thus, the resulting model defines a GMRF w.r.t the $2L$-hop graph of $\bar{\mathcal{G}}$.

# 3 Spatiotemporal DGMRFs

To extend the DGMRF framework to spatiotemporal systems, we first formulate a dynamical prior in terms of a GMRF that encodes prior knowledge in the form of spatial and temporal independence assumptions. We then parameterize this prior using simple spatial and temporal layers, which results in a flexible model architecture that facilitates efficient learning and principled Bayesian inference in high-dimensional dynamical systems, scaling favorably compared to Kalman filter based methods.

## 3.1 Graph-structured dynamical prior

Consider a dynamical system for which the state evolution is well described by Eq. (2). We say that this process is graph-structured if each dimension $d_i$ of the system state $\mathbf{x}_k$ can be associated with a node $i \in \mathcal{V}$ in a multigraph $\mathcal{G} = (\mathcal{V}, \mathcal{E}_{\text{spatial}}, \mathcal{E}_{\text{temporal}})$, where the set of undirected spatial edges $\mathcal{E}_{\text{spatial}} \subseteq \mathcal{V} \times \mathcal{V}$ defines the sparsity pattern of noise precision (inverse covariance) matrices

$$(\mathbf{Q}_k)_{ij} \neq 0 \iff (\mathbf{Q}_k)_{ji} \neq 0 \iff (j, i) \in \mathcal{E}_{\text{spatial}} \qquad \forall k \in \{0, \dots, K\}, \quad (5)$$

and the set of directed temporal edges $\mathcal{E}_{\text{temporal}} \subseteq \mathcal{V} \times \mathcal{V}$ defines the sparsity pattern of the state transition matrices

$$(\mathbf{F}_k)_{ij} \neq 0 \iff (j, i) \in \mathcal{E}_{\text{temporal}} \qquad \forall k \in \{1, \dots, K\}. \quad (6)$$

This defines a DBN over $\mathbf{x}_{0:K}$ encoding conditional independencies of the form

$$x_{k,i} \perp\!\!\!\perp \mathbf{x}_{k-1, \mathcal{V} \backslash n_{\text{temporal}}(i)} \mid \mathbf{x}_{k-1, n_{\text{temporal}}(i)} \quad \text{and} \quad x_{k,i} \perp\!\!\!\perp \mathbf{x}_{l,\mathcal{V}} \mid \mathbf{x}_{k-1, n_{\text{temporal}}(i)} \; \forall l < k-1, \quad (7)$$

where $n_{\text{temporal}}(i)$ denotes the set of neighbors $j$ of $i$ for which $(j, i) \in \mathcal{E}_{\text{temporal}}$. Intuitively, $\mathcal{E}_{\text{temporal}}$ represent causal effects over time, while $\mathcal{E}_{\text{spatial}}$ represent the structure of random effects that are not captured by the transition model. To define the graph structure, prior knowledge about, for example, the physical system structure or underlying causal mechanisms can be exploited.

### 3.1.1 Joint distribution

The graph-structured dynamical prior induces a multivariate Gaussian prior $\mathcal{N}(\boldsymbol{\mu}, \boldsymbol{\Omega}^{-1})$ on $\mathbf{x}$ with sparse precision matrix $\boldsymbol{\Omega} \in \mathbb{R}^{(K+1)N \times (K+1)N}$. Importantly, $\boldsymbol{\Omega}$ can be shown to factorize as $\mathbf{F}^T \mathbf{Q} \mathbf{F}$ with block diagonal matrix $\mathbf{Q}$ and unit lower block bidiagonal matrix $\mathbf{F}$ defined as

$$\mathbf{Q} = \text{diag}(\mathbf{Q}_0, \mathbf{Q}_1, \dots, \mathbf{Q}_K), \qquad \mathbf{F} := \begin{bmatrix} \mathbf{I} & & & \\ -\mathbf{F}_1 & \mathbf{I} & & \\ & \dots & \dots & \\ & & -\mathbf{F}_K & \mathbf{I} \end{bmatrix}, \quad (8)$$

where empty positions represent zero-blocks. Further, while the mean $\boldsymbol{\mu} \in \mathbb{R}^{(K+1)N}$ needs to be computed iteratively as $\boldsymbol{\mu}_k = \mathbf{F}_k \boldsymbol{\mu}_{k-1} + \mathbf{c}_k$, the information vector $\boldsymbol{\eta} = \boldsymbol{\Omega}\boldsymbol{\mu}$ can be expressed compactly as

$$\boldsymbol{\eta} = \mathbf{F}^T \mathbf{Q} \mathbf{F} \boldsymbol{\mu} = \mathbf{F}^T \mathbf{Q} \mathbf{c} \quad (9)$$

with $\mathbf{c} = [\boldsymbol{\mu}_0, \mathbf{c}_1, \dots, \mathbf{c}_K] \in \mathbb{R}^{(K+1)N}$. See Appendix A.1 for detailed derivations.

### 3.1.2 DGMRF formulation

We now reformulate $\mathbf{x} \sim \mathcal{N}(\boldsymbol{\mu}, \boldsymbol{\Omega}^{-1})$ as a DGMRF. Importantly, in contrast to [50, 43], the graph-structured dynamical prior allows us to impose additional structure, reflecting the spatiotemporal nature of the underlying system. In particular, since $\mathbf{Q}_0, \mathbf{Q}_1, \dots, \mathbf{Q}_K$ are symmetric positive definite, we can factorize $\mathbf{Q} = \mathbf{S}^T \mathbf{S}$ with symmetric block-diagonal matrix $\mathbf{S}$, and thus $\boldsymbol{\Omega} = \mathbf{F}^T \mathbf{S}^T \mathbf{S} \mathbf{F}$. Together with Eq. (9), this results in a GMRF defined by

$$\mathbf{G}_\theta := \mathbf{S}\mathbf{F} \quad \text{and} \quad \mathbf{b}_\theta := -\mathbf{G}_\theta \boldsymbol{\mu} = -\mathbf{S}\mathbf{c} \quad (10)$$

Finally, we separate $\mathbf{g}_\theta$ into a temporal map $\mathbf{f}_\theta : \mathbb{R}^{(K+1)N} \to \mathbb{R}^{(K+1)N}$ and a spatial map $\mathbf{s}_\theta : \mathbb{R}^{(K+1)N} \to \mathbb{R}^{(K+1)N}$ defined as

$$\mathbf{f}_\theta(\mathbf{x}) := \mathbf{F}\mathbf{x} + \mathbf{b}_f = \mathbf{h} \quad \text{and} \quad \mathbf{s}_\theta(\mathbf{h}) := \mathbf{S}\mathbf{h} + \mathbf{b}_s = \mathbf{z}. \quad (11)$$

Note that this results in an overall bias $\mathbf{b}_\theta = \mathbf{S}\mathbf{b}_f + \mathbf{b}_s$ and thus $\mathbf{c} = -(\mathbf{b}_f + \mathbf{S}^{-1}\mathbf{b}_s)$, which allows for modelling long-range spatial dependencies in $\boldsymbol{\mu}_0$ and $\mathbf{c}_1, \dots, \mathbf{c}_K$. The combined transformation $\mathbf{z} = (\mathbf{s}_\theta \circ \mathbf{f}_\theta)(\mathbf{x})$ essentially defines a standard two-layer DGMRF, which describes a graph-structured dynamical system if the parameters $\boldsymbol{\theta} = (\mathbf{b}_s, \mathbf{b}_f, \mathbf{F}_1, \dots, \mathbf{F}_K, \mathbf{Q}_0, \dots, \mathbf{Q}_K)$ are subject to sparsity constraints (5) and (6).

## 3.2 Parameterization

Learning the unknown parameters $\boldsymbol{\theta}$ directly from a single sequence $\mathbf{y}_{0:K}$ will result in a highly over-parameterized model that is unsuitable for the problem of interest. In addition, careful parameterization of $\mathbf{g}_\theta$ can greatly reduce the computational complexity of the transformation and the associated log-determinant computations required during learning [50, 43]. To achieve a good trade-off between data-efficiency, expressivity and scalability, we define both $\mathbf{s}_\theta$ and $\mathbf{f}_\theta$ in terms of simple layers with few parameters and convenient computational properties.

### 3.2.1 Spatial layer(s)

To enable fast log-determinant computations for arbitrary graph structures, we follow [43] and parameterize $\mathbf{s}_\theta$ in terms of $K+1$ independent DGMRFs $\mathbf{z}_k = \mathbf{S}_k \mathbf{h}_k + (\mathbf{b}_s)_k$, each defining a GMRF w.r.t $\mathcal{G}_{\text{spatial}} = (\mathcal{V}, \mathcal{E}_{\text{spatial}})$. Note that due to the multi-layer construction (see Section 2.4), $\mathcal{G}_{\text{spatial}}$ is implicitly defined by the base graph $\bar{\mathcal{G}}_{\text{spatial}}$ and the number of layers $L$, with special case $\mathcal{G}_{\text{spatial}} = \bar{\mathcal{G}}_{\text{spatial}}$ if $L_{\text{spatial}} = 1$.

### 3.2.2 Temporal layer(s)

Compared to $\mathbf{s}_\theta$, the definition of $\mathbf{f}_\theta$ is much less constrained as it does not affect the log-determinant computation (see Section 3.3.1), and should hence incorporate as much domain knowledge into transition matrices $\mathbf{F}_k$ as possible. This could be, for example, based on conservation laws or knowledge about relevant covariates. To increase the flexibility in cases where the dynamics are unknown or involve long-range dependencies, each $\mathbf{F}_k$ can again be decomposed into simpler layers $\mathbf{F}_k^{(L_{\text{temporal}})} \cdots \mathbf{F}_k^{(1)}$, each defined according to a temporal base graph $\bar{\mathcal{G}}_{\text{temporal}}$. Table 1 provides several example layers to give an idea of what is possible.

Note that any function can be used to define the entries of $\mathbf{F}_k^{(l)}$, including neural networks taking available covariates or node/edge features as inputs [46]. The associated parameters can simply be included in $\boldsymbol{\theta}$. Similarly, non-linear dynamics can be approximated either through linearization akin to the Extended Kalman Filter [37], or through neural linearization [19, 7]. We, however, recommend sharing parameters where appropriate to avoid overparameterization.

Table 1: Examples of linear transition layers $\mathbf{F}_k^{(l)}$. The adjacency matrix $\mathbf{A}$ can be symmetric or asymmetric, weighted or unweighted.

|  | layer definition | properties of $\mathcal{G}_{\text{temporal}}$ |
|---|---|---|
| AR process | $\mathbf{F}_k^{(l)} = \lambda_{k,l}\mathbf{I}$ | self-edges only |
| Diffusion | $\mathbf{F}_k^{(l)} = \lambda_{k,l}\mathbf{I} + \omega_{k,l}(\mathbf{A} - \mathbf{D})$ | bidirected (symmetric $\mathbf{A}$) |
| Directed flow | $\mathbf{F}_k^{(l)} = \lambda_{k,l}\mathbf{I} + \omega_{k,l}(\mathbf{A} - \mathbf{D}_{\text{out}}) + \zeta_{k,l}(\mathbf{A}^T - \mathbf{D}_{\text{in}})$ | directed |
| Advection | $\left(\mathbf{F}_k^{(l)}\right)_{ij} = -\frac{1}{2}w_{ij}\mathbf{n}_{ij}^T\mathbf{v}_l$ 
 $\left(\mathbf{F}_k^{(l)}\right)_{ii} = 1 - \sum_{j \in n(i)}\left(\mathbf{F}_k^{(l)}\right)_{ij}$ | discretized $\mathbb{R}^d$ (e.g. triangulation), 
 edge weights $w_{ij}$ and normals $\mathbf{n}_{ij}$ |
| Neural network | $\left(\mathbf{F}_k^{(l)}\right)_{ij} = f_{\text{NN}}(\mathbf{u}_i, \mathbf{u}_j, \mathbf{e}_{ij})$ | node and edge features $\mathbf{u}_i, \mathbf{e}_{ij}$ |

**Higher order Markov processes** It is straight forward to extend $\mathbf{f}_\theta$ to describe a $p$-th order Markov process

$$\mathbf{x}_k = \sum_{\tau=1}^{p} \mathbf{F}_{k,\tau}\mathbf{x}_{k-\tau} + \mathbf{c}_k + \boldsymbol{\epsilon}_k, \quad \boldsymbol{\epsilon} \sim \mathcal{N}(\mathbf{0}, \mathbf{Q}_k^{-1}), \quad \mathbf{x}_0 \sim \mathcal{N}(\boldsymbol{\mu}_0, \mathbf{Q}_0^{-1}), \quad (12)$$

where $\mathbf{x}_{-\tau} = \mathbf{0}$ for $\tau = 1, \ldots, p$. This can be done by introducing edges $\mathcal{E}_{\text{temporal}(2)}, \ldots, \mathcal{E}_{\text{temporal}(p)}$ and adding the corresponding higher-order transition matrices $(\mathbf{F}_{\tau,\tau}, \ldots, \mathbf{F}_{K,\tau})$ to the $\tau$-th lower block diagonal of $\mathbf{F}$ (see Appendix A.2 for derivations).

## 3.3 Learning and inference

Although the marginal likelihood $p(\mathbf{y} \mid \boldsymbol{\theta})$ is available in closed form (see Section 2.3), maximum likelihood parameter estimation becomes computationally prohibitive in high dimensional settings

[50]. Instead, a variational approximation is used to learn parameters $\hat{\boldsymbol{\theta}}$ for large-scale DGMRFs. Then, the conjugate gradient method allows to efficiently compute the exact posterior mean and to draw samples from $p(\mathbf{x} \mid \mathbf{y}, \hat{\boldsymbol{\theta}})$.

### 3.3.1 Scalable parameter estimation

Given a variational distribution $q_\phi(\mathbf{x}) = \mathcal{N}(\boldsymbol{\nu}_\phi, \boldsymbol{\Lambda}_\phi)$, the parameters $\{\boldsymbol{\theta}, \boldsymbol{\phi}\}$ are optimized jointly by maximizing the Evidence Lower Bound (ELBO)

$$\mathcal{L}(\mathbf{y}_{0:K}, \boldsymbol{\theta}, \boldsymbol{\phi}) = \mathbb{E}_{q_\phi(\mathbf{x})}\left[\log p_\theta(\mathbf{x}) + \sum_{k=0}^{K} \log p(\mathbf{y}_k \mid \mathbf{x}_k)\right] + H\left[q_\phi(\mathbf{x})\right] \tag{13}$$

$$= -\frac{1}{2}\mathbb{E}_{q_\phi(\mathbf{x})}\left[\mathbf{g}_\theta(\mathbf{x})^T\mathbf{g}_\theta(\mathbf{x}) + \sum_{k=0}^{K}(\mathbf{y}_k - \mathbf{H}_k\mathbf{x}_k)^T\mathbf{R}_k^{-1}(\mathbf{y}_k - \mathbf{H}_k\mathbf{x}_k)\right] \tag{14}$$

$$+ \log|\det(\mathbf{G}_\theta)| + \frac{1}{2}\log|\det(\boldsymbol{\Lambda}_\phi)| - \frac{1}{2}\sum_{k=0}^{K}\log|\det(\mathbf{R}_k)| + \text{const}, \tag{15}$$

using stochastic gradient descent, where the expectation is replaced by a Monte-Carlo estimate based on samples $\hat{\mathbf{x}} \sim q_\phi$.

**Variational distribution** To facilitate efficient log-determinant computations and sampling via the reparameterization trick [33], we follow [43] and define $q_\phi$ as an affine transformation $\mathbf{x} = \mathbf{P}_\phi\mathbf{z} + \boldsymbol{\nu}_\phi$ with $\mathbf{z} \sim \mathcal{N}(\mathbf{0}, \mathbf{I})$, resulting in $\boldsymbol{\Lambda}_\phi = \mathbf{P}_\phi(\mathbf{P}_\phi)^T$. Then, $\mathbf{P}_\phi$ can be defined as a block-diagonal matrix $\text{diag}(\mathbf{P}_0, \ldots, \mathbf{P}_K)$ with $\mathbf{P}_k = \text{diag}(\boldsymbol{\rho}_k)\tilde{\mathbf{S}}_k\text{diag}(\boldsymbol{\psi}_k)$, where $\boldsymbol{\rho}_k, \boldsymbol{\psi}_k \in \mathbb{R}_+^N$ are variational parameters and $\tilde{\mathbf{S}}_k$ is based on spatial DGMRF layers as discussed in Section 3.2.1. To relax the temporal independence assumptions in $\boldsymbol{\Lambda}_\phi$, we propose an extension $\mathbf{P}_\phi = \text{diag}(\mathbf{P}_0, \cdots, \mathbf{P}_K)\tilde{\mathbf{F}}$, where $\tilde{\mathbf{F}}$ has a similar structure to $\mathbf{F}$ (see Eq. (8)). Again, the design of $\tilde{\mathbf{F}}_k$ is highly flexible as it does not enter the log-determinant computation.

**Log-determinant computations** [50, 43] proposed specific lattice and graph layers for which the associated log-determinants $\log|\det(\mathbf{G}_\theta)|$ are computationally scalable. Conveniently, the spatiotemporal case does not require a new type of layer. Instead, we can build directly on top of existing spatial layers. Specifically, using Eq. (8)&(10), the log-determinant $\log|\det(\mathbf{G}_\theta)|$ simplifies to

$$\log|\det(\mathbf{SF})| \overset{(i)}{=} \log|\det(\mathbf{S})| \overset{(ii)}{=} \sum_{k=0}^{K}\log|\det(\mathbf{S}_k)|, \tag{16}$$

where $(i)$ follows from $\mathbf{S}$ being block-diagonal and $(ii)$ follows from $\mathbf{F}$ being unit-lower triangular with $\det(\mathbf{F}) = 1$. Note that each $\mathbf{S}_k$ is defined by a spatial DGMRF of dimension $N$ (see Section 3.2.1), for which efficient log-determinant methods have been developed. The same arguments hold for the variational distribution proposed above. Finally, due to the diagonality assumptions on $\mathbf{R}_t$ (see Section 2.1), $\frac{1}{2}\log|\det(\mathbf{R}_k)|$ simplifies to $\sum_{i=1}^{M_k}\log(\sigma_i)$.

**Computational complexity** During training, $\log|\det(\mathbf{G}_\theta)|$ and $\log|\det(\boldsymbol{\Lambda}_\phi)|$ can be computed in $\mathcal{O}(KNL_{\text{spatial}})$ using Eq. (16) together with the methods proposed in [43]. Note that no complexity is added when introducing conditional dependencies between time steps. The necessary preprocessing steps, i.e. computing eigenvalues or traces, only need to be performed for the spatial base graph and are thus independent of the number of transitions $K$. Finally, assuming an average of $d_{\text{spatial}}$ and $d_{\text{temporal}}$ edges per node in $\bar{\mathcal{E}}_{\text{spatial}}$ and $\bar{\mathcal{E}}_{\text{temporal}}$, the transformations $\mathbf{s}_\theta$ and $\mathbf{f}_\theta$ scale linearly with the number of nodes $N$. In particular, $\mathbf{g}_\theta(\mathbf{x}) = (\mathbf{s}_\theta \circ \mathbf{f}_\theta)(\mathbf{x})$ can be computed in $\mathcal{O}(KNd_{\text{spatial}}L_{\text{spatial}} + KNd_{\text{temporal}}L_{\text{temporal}})$, which dominates the computational complexity of the training loop. In addition, we can leverage massively parallel GPU computations to speed up this process even further.

### 3.3.2 Exact inference with conjugate gradients

As discussed in Section 2.3, the conjugate gradient method can be used to iteratively compute the posterior mean and obtain Monte Carlo estimates of marginal variances. Importantly, each iteration

is dominated by a single matrix-vector multiplication of the form

$$\mathbf{\Omega}^+\mathbf{x} = \mathbf{F}^T\mathbf{S}^T\mathbf{S}\mathbf{F}\mathbf{x} + \mathbf{H}^T\mathbf{R}^{-1}\mathbf{H}\mathbf{x}. \tag{17}$$

This amounts to a series of sparse matrix-vector multiplications with total computational complexity $\mathcal{O}(KNd_{\text{spatial}}L_{\text{spatial}} + KNd_{\text{temporal}}L_{\text{temporal}})$, which is again linear in the number of nodes, time steps, and layers, and can be implemented in parallel on a GPU.

## 4    Experiments

We implemented ST-DGMRF in Pytorch and Pytorch Geometric, and conducted experiments on a consumer-grade GPU, leveraging parallel computations in both spatial and temporal layers. In all experiments, we optimize parameters for $10\,000$ iterations using Adam [32] with learning rate $0.01$, and draw 100 posterior samples to estimate marginal variances. Unless specified otherwise, we use $L_{\text{spatial}} = 2$, $L_{\text{temporal}} = 4$ and $p = 1$, and define the variational distribution based on one spatial layer and one temporal *diffusion* layer. Additional details on our experiments are provided in Appendix B.

### 4.1    Advection-diffusion process

We start with a synthetic dataset for which we have access to both the ground truth posterior distribution and transition matrix. The dataset consists of $K = 20$ system states that are sampled from an ST-DGMRF with time-invariant transition matrix $\mathbf{F}_{\text{adv-diff}}$. This matrix is defined according to the third-order Taylor approximation of an advection-diffusion process with constant velocity and diffusion parameters, discretized on a $30 \times 30$ lattice with periodic boundary conditions. From the sampled state trajectory, we generate observations with varying amount of missing data by removing pixels within a square mask of width $w$ for 10 consecutive time steps and adding noise with $\sigma = 0.01$. For all experiments, we use the masked pixels as test set, and 10% of the observed pixels as validation set for hyperparameter tuning.

Two ST-DGMRF variants are considered: (1) using advection-diffusion matrices $\mathbf{F}_k^{(l)}$ defined based on the first-order Taylor approximation of the ground-truth dynamics, with trainable diffusion and velocity parameters, and (2) replacing parts of these advection-diffusion matrices with small neural networks, taking the edge unit vector $\mathbf{n}_{ij}$ pointing from pixel $i$ to $j$ as input. In both cases, $\bar{\mathcal{G}}_{\text{temporal}}$ and $\bar{\mathcal{G}}_{\text{spatial}}$ are defined as the 4-nearest neighbor graph. Further, as the underlying dynamics are time-invariant, we enforce $\mathbf{F}_k^{(l)} = \mathbf{F}_{k+1}^{(l)}$ and $\mathbf{S}_k^{(l)} = \mathbf{S}_{k+1}^{(l)}$ $\forall k \geq 1$.

**Baselines**    We compare our approach to a range of baselines accounting for varying degrees of spatial and/or temporal dependencies. In particular, we consider the original DGMRF applied to all time steps independently, an ARMA state-space model assuming spatial independence among the state variables, and a spatiotemporal AR state-space model (ST-AR) with spatially correlated error terms $\epsilon_k$ for which an unconstrained covariance matrix $\mathbf{Q}^{-1}$ is estimated using the EM algorithm. For both ARMA and ST-AR, we use the standard Kalman smoother [47] to obtain posterior estimates. Additionally, we consider two Ensemble Kalman Smoother (EnKS) variants, one with an advection-diffusion transition model matching the

Table 2: Performance on the advection-diffusion data with $w = 9$. We report the mean over 5 runs with different random seeds.

|  | $\text{RMSE}_\mu$ | $\text{RMSE}_\sigma$ | CRPS |
|---|---|---|---|
| ARMA | 2.3054 | 0.6812 | 1.7064 |
| ST-AR | 1.4595 | 1.9216 | 0.9707 |
| DGMRF | 0.5901 | 0.3808 | 0.3495 |
| EnKS |  |  |  |
| *true dynamics* | 0.0661 | 0.0046 | 0.1027 |
| *estimated dynamics* | 0.1654 | **0.0039** | 0.1434 |
| ST-DGMRF (ours) |  |  |  |
| *advection-diffusion* | **0.0526** | 0.1146 | **0.0726** |
| *neural network* | 0.0854 | 0.1402 | 0.0839 |

true data-generating process, and one using state augmentation to estimate the velocity and diffusion parameters jointly with the system states. Note that, in contrast to the ST-DGMRF approach, we consider initial and transition noise parameters to be fixed in order to avoid divergence of the EnKS.

**Performance evaluation**    We evaluate the estimated posterior mean and marginal standard deviations in terms of the root-mean-square-error ($\text{RMSE}_\mu$ and $\text{RMSE}_\sigma$) with respect to the ground truth posterior. In addition, we use the mean negative continuous ranked probability score (CRPS) [22] to evaluate the predictive distribution with respect to the masked out data. Table 2 shows that, by exploiting the spatiotemporal structure of the process, our ST-DGMRF variants provide much more

accurate estimates than the purely spatial DGMRF, the purely temporal ARMA model, and the ST-AR model with highly simplified transitions and unstructured noise terms. As expected, the two EnKS variants with fixed noise parameters provide the most accurate uncertainty estimates. However, the CRPS scores indicate that overall our ST-DGMRF approach results in better calibrated posterior distributions. More detailed results are reported in Appendix C.

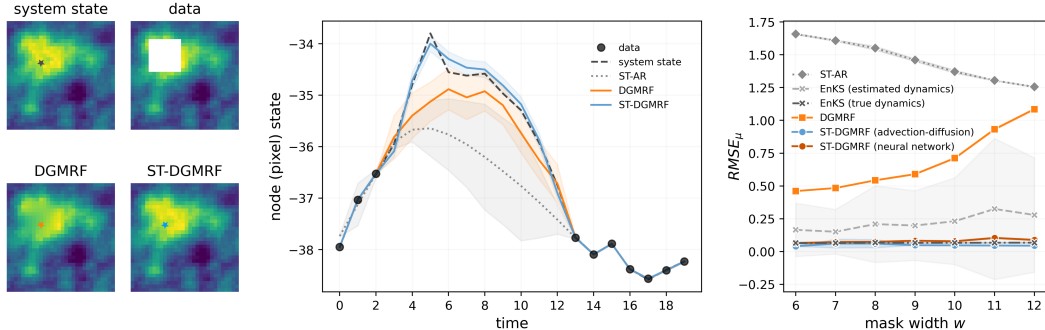

Figure 2: Left: snapshot of the advection-diffusion data at time $k$=8, and reconstructions by the time-independent DGMRF and our ST-DGMRF with advection-diffusion $\mathbf{F}_k$. Center: corresponding time series for a single pixel (marked on the left). Shaded areas represent posterior mean $\pm$ std of a single run. Right: $\text{RMSE}_\mu$ as a function of the mask width (mean $\pm$ std over 5 runs).

**Increasing mask size** To evaluate the robustness to missing data, we analyze how posterior estimates change with increasing mask size. Figure 2 (right) shows the effect on $\text{RMSE}_\mu$ when varying $w$ from 6 to 12. Clearly, the purely spatial DGMRF suffers the most from expanding the unobserved region. In contrast, the ST-DGMRF variants continue to provide accurate state estimates that are on par with the EnKS using ground truth dynamics. Note that the decreasing errors for ST-AR can be attributed to the changing set of pixels used for evaluation.

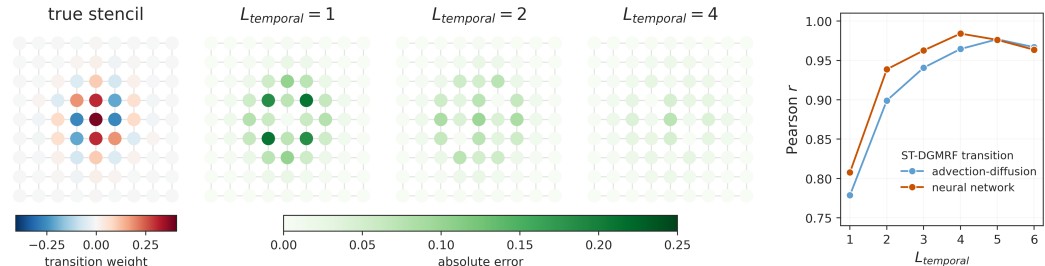

Figure 3: Stencil comparison. Center: absolute error between true and learned transition weights for the neural network based ST-DGMRF. Right: Pearson correlation for increasing temporal depth.

**Evaluation of learned transition models** Finally, we assess how well the temporal ST-DGMRF layers can approximate the true data-generating dynamics, given varying levels of complexity in the transition model. For this purpose, we extract the stencil, i.e. the weights assigned to nearby pixels, from the ground truth and learned transition matrices and compare them in terms of absolute error and Pearson correlation. We find that the learned transition weights converge rapidly towards the true weights as the number of temporal layers increases (see Figure 3). For small $L_{\text{temporal}}$, the neural network based layers show a stronger agreement with the true dynamics, suggesting that some flexibility in the temporal layers helps to compensate for simplifications in the transition structure.

## 4.2 Air quality data

To test our method on a real world system exhibiting more complex dynamics and graph structures, we conduct experiments on an air quality dataset obtained from [59]. The dataset contains hourly PM2.5 measurements from 246 sensors distributed around Beijing, China, covering a time period of $K = 400$ hours. To ensure that predicted PM2.5 concentrations are non-negative, we model both system states and observations in log-space. The spatial and temporal base graphs are defined

based on the Delaunay triangulation, with edge weights proportional to the inverse distance between sensors. To mimic a realistic scenario of local network failures, we randomly choose 10 time points $t_k$ and mask out all data points within a spatial block containing 50% of all sensors, for time steps $t_k, \ldots, t_k + 20$ (see Figure 4). As before, we use the masked data for model evaluation, and use 10% of the remaining data as validation set. The observation noise is assumed to be uniform with $\sigma = 0.01$.

The spatiotemporal distribution of particulate matter is strongly influenced by atmospheric processes that transport and diffuse emitted particles. To incorporate this knowledge into the transition model, we extract temperature and wind conditions for all sensors and time points from the ERA5 reanalysis dataset [25] and feed them together with static graph features into a set of neural networks which transform them into spatially and temporally varying bias and velocity parameters. The matrices $\mathbf{F}_k^{(l)}$ are then formed in the same way as in Section 4.1. In addition, we consider a simplified diffusion transition model (see Table 1) which cannot capture any directional transport processes.

**Results** Next to the baselines introduced in Section 4.1, we consider a multi-layer perceptron (MLP) mapping local weather features to the corresponding log-transformed PM2.5 concentration. Following [50], we also include weather features into the spatial DGMRF model by adding linear effects to the measurement model. Table 3 reports the resulting CRPS and the RMSE with respect to the masked out data. Clearly, the neural network based ST-DGMRF, accounting for time-varying directional transport processes, provides the most accurate state estimates. However, even with highly simplified *diffusion* transitions our approach provides better estimates than the considered baselines. We attribute this to the expressive DGMRF noise terms which can capture complex error structures

Table 3: Performance on the air quality data. We report the mean over 5 runs with different random seeds.

|  | p | RMSE | CRPS |
|---|---|---|---|
| ARMA |  | 0.6820 | 0.3625 |
| ST-AR |  | 0.7350 | 0.4261 |
| DGMRF |  | 0.7456 | 0.4037 |
| MLP |  | 0.8038 | – |
| ST-DGMRF (ours) |  |  |  |
| *diffusion* | 1 | 0.6190 | 0.3258 |
| *diffusion* | 2 | 0.5928 | 0.3161 |
| *neural network* | 1 | 0.5853 | 0.3092 |
| *neural network* | 2 | **0.5565** | **0.2925** |

resulting from inaccuracies in the transition model. Increasing the Markov order from $p = 1$ to $2$ clearly improves the resulting posterior estimates for both ST-DGMRF variants, reflecting the complexity of the modeled process. A similar trend is observed as we increase $L_{\text{temporal}}$ (see Appendix C).

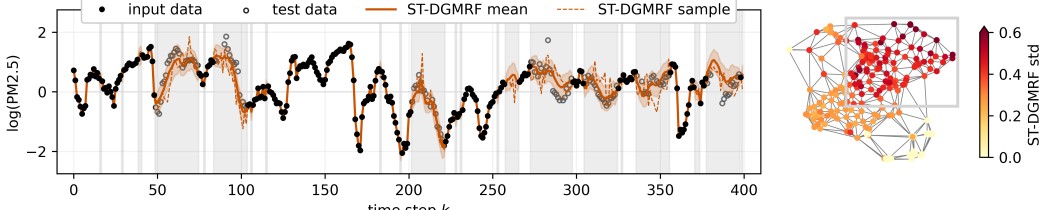

Figure 4: ST-DGMRF posterior estimates (*neural network* $\mathbf{F}_k$ and $p = 2$). Left: time series of log-transformed and normalized PM2.5 levels for a masked sensor. Light red areas represent posterior mean $\pm$ std of a single run. Unobserved time points are marked with gray bars. Right: Marginal std for all sensors at $k = 100$. Masked out sensors (gray box) feature higher uncertainties.

## 5 Conclusion

We have presented ST-DGMRF, an extension to Deep Gaussian Markov Random Fields for inference in spatiotemporal dynamical systems with partial and noisy observations, (partially) unknown dynamics, and limited historical data. Our reformulation of graph-structured state-space models as multi-layer space-time GMRFs enables computationally efficient learning and inference even in high-dimensional settings, scaling linear w.r.t. the number of both time steps and state variables. Empirically, ST-DGMRF provides more accurate posterior estimates than other scalable approaches relying on simplifications of the dependency structure. While our approach relies on linear Gaussian assumptions, which can be restrictive for real systems, we find that expressive DGMRF noise terms can compensate (to a certain extend) for approximations in the transition model. In the future, non-linearities between the layers could be explored as discussed in [50]. Further, we see potential in defining more flexible time-varying transition matrices based on a hierarchy of latent variables that again follow a ST-DGMRF.

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

# Appendices

Appendix A provides derivations supporting Section 3 in the main paper. In Appendix B, we explain our experimental setup, including dataset preparation and model implementation, in more detail. Finally, Appendix C provides additional results supporting our claims regarding the scalability of our method, together with additional results from the experiments presented in Section 4.

## A ST-DGMRF derivations

In this section we provide detailed derivations of the ST-DGMRF joint distribution, for both first-order transition models (Section A.1) and higher-order transition models (Section A.2).

### A.1 Joint distribution

The LDS (see Section 2.2 and 3.1 in the main paper) defines a joint distribution over system states $\mathbf{x}_{0:K}$ that factorizes as

$$p(\mathbf{x}_{0:K}) = \mathcal{N}(\mathbf{x}_0 \mid \boldsymbol{\mu}_0, \mathbf{Q}_0^{-1}) \prod_{k=1}^{K} \mathcal{N}(\mathbf{x}_k \mid \mathbf{F}_k \mathbf{x}_{k-1} + \mathbf{c}_k, \mathbf{Q}_k^{-1}), \tag{18}$$

with $\mathbf{x}_k, \boldsymbol{\mu}_0, \mathbf{c}_k \in \mathbb{R}^N$ and $\mathbf{F}_k, \mathbf{Q}_k \in \mathbb{R}^{N \times N}$. As a product of Gaussian distributions, $p(\mathbf{x}_{0:K})$ can be written as a joint Gaussian $\mathcal{N}(\boldsymbol{\mu}, \boldsymbol{\Omega}^{-1})$ with mean $\boldsymbol{\mu} \in \mathbb{R}^{(K+1)N}$ and precision (inverse covariance) matrix $\boldsymbol{\Omega} \in \mathbb{R}^{(K+1)N \times (K+1)N}$. Here, we derive expressions for $\boldsymbol{\mu}$ and $\boldsymbol{\Omega}$ in terms of $\boldsymbol{\mu}_0, \mathbf{c}_k, \mathbf{F}_k, \mathbf{Q}_k$.

First, note that Eq. (18) can be written as a set of linear equations

$$\begin{aligned}
\mathbf{x}_0 &= \boldsymbol{\mu}_0 + \boldsymbol{\epsilon}_0 & \boldsymbol{\epsilon}_0 &\sim \mathcal{N}(\mathbf{0}, \mathbf{Q}_0^{-1}) \\
\mathbf{x}_1 &= \mathbf{F}_1 \mathbf{x}_0 + \mathbf{c}_1 + \boldsymbol{\epsilon}_1 & \boldsymbol{\epsilon}_1 &\sim \mathcal{N}(\mathbf{0}, \mathbf{Q}_1^{-1}) \\
\mathbf{x}_2 &= \mathbf{F}_2 \mathbf{x}_1 + \mathbf{c}_2 + \boldsymbol{\epsilon}_2 & \boldsymbol{\epsilon}_2 &\sim \mathcal{N}(\mathbf{0}, \mathbf{Q}_2^{-1}) \\
&\cdots \\
\mathbf{x}_K &= \mathbf{F}_K \mathbf{x}_{K-1} + \mathbf{c}_K + \boldsymbol{\epsilon}_K & \boldsymbol{\epsilon}_K &\sim \mathcal{N}(\mathbf{0}, \mathbf{Q}_K^{-1}).
\end{aligned}$$

Moving all $\mathbf{x}_k$-terms to the left-hand side, we can rewrite this as a matrix-vector multiplication

$$\underbrace{\begin{bmatrix} \mathbf{I} & & & & \\ -\mathbf{F}_1 & \mathbf{I} & & & \\ & -\mathbf{F}_2 & \mathbf{I} & & \\ & & \cdots & \cdots & \\ & & & -\mathbf{F}_K & \mathbf{I} \end{bmatrix}}_{=\mathbf{F}} \cdot \underbrace{\begin{bmatrix} \mathbf{x}_0 \\ \mathbf{x}_1 \\ \mathbf{x}_2 \\ \vdots \\ \mathbf{x}_K \end{bmatrix}}_{=\mathbf{x}} = \underbrace{\begin{bmatrix} \boldsymbol{\mu}_0 \\ \mathbf{c}_1 \\ \mathbf{c}_2 \\ \vdots \\ \mathbf{c}_K \end{bmatrix}}_{=\mathbf{c}} + \underbrace{\begin{bmatrix} \boldsymbol{\epsilon}_0 \\ \boldsymbol{\epsilon}_1 \\ \boldsymbol{\epsilon}_2 \\ \vdots \\ \boldsymbol{\epsilon}_K \end{bmatrix}}_{=\boldsymbol{\epsilon}}, \tag{19}$$

with block-matrix $\mathbf{F} \in \mathbb{R}^{(K+1)N \times (K+1)N}$ and vectorized $\mathbf{x} = \text{vec}(\mathbf{x}_0, \ldots, \mathbf{x}_K) \in \mathbb{R}^{(K+1)N}$, $\mathbf{c} = \text{vec}(\boldsymbol{\mu}_0, \mathbf{c}_1, \ldots, \mathbf{c}_K) \in \mathbb{R}^{(K+1)N}$ and $\boldsymbol{\epsilon} = \text{vec}(\boldsymbol{\epsilon}_0, \ldots, \boldsymbol{\epsilon}_K) \in \mathbb{R}^{(K+1)N}$. Empty positions in $\mathbf{F}$ represent zero-blocks.

Now, we can express $\mathbf{x}$ as an affine transformation of $\boldsymbol{\epsilon}$

$$\mathbf{x} = \mathbf{F}^{-1}\mathbf{c} + \mathbf{F}^{-1}\boldsymbol{\epsilon}, \tag{20}$$

where $\mathbf{F}^{-1}$ exists because $\det(\mathbf{F}) = 1$. Since $\boldsymbol{\epsilon}$ is distributed as $\boldsymbol{\epsilon} \sim \mathcal{N}(\mathbf{0}, \mathbf{Q}^{-1})$ with $\mathbf{Q} = \text{diag}(\mathbf{Q}_0, \mathbf{Q}_1, \ldots, \mathbf{Q}_K)$, and $\mathbf{c}$ is deterministic, we can use the affine property of Gaussian distributions to obtain the joint distribution

$$\mathbf{x} \sim \mathcal{N}(\mathbf{F}^{-1}\mathbf{c}, \mathbf{F}^{-1}\mathbf{Q}^{-1}\mathbf{F}^{-T}). \tag{21}$$

Thus, the joint precision matrix $\boldsymbol{\Omega}$ factorizes as

$$\boldsymbol{\Omega} = (\mathbf{F}^{-1}\mathbf{Q}^{-1}\mathbf{F}^{-T})^{-1} = \mathbf{F}^T \mathbf{Q} \mathbf{F} \tag{22}$$

and has a block-tridiagonal structure

$$
\Omega =
\begin{bmatrix}
\mathbf{Q}_0 + \mathbf{F}_1^T \mathbf{Q}_1 \mathbf{F}_1 & -\mathbf{F}_1^T \mathbf{Q}_1 \\
-\mathbf{Q}_1 \mathbf{F}_1 & \mathbf{Q}_1 + \mathbf{F}_2^T \mathbf{Q}_2 \mathbf{F}_2 & -\mathbf{F}_2^T \mathbf{Q}_2 \\
& \cdots & \cdots & \cdots \\
& & -\mathbf{Q}_{K-1} \mathbf{F}_{K-1} & \mathbf{Q}_{K-1} + \mathbf{F}_K^T \mathbf{Q}_K \mathbf{F}_T & -\mathbf{F}_K^T \mathbf{Q}_K \\
& & & -\mathbf{Q}_K \mathbf{F}_K & \mathbf{Q}_K
\end{bmatrix} .
\tag{23}
$$

Note that for matrix-vector multiplications of the form $\Omega \mathbf{x}$, the sparse structure of $\mathbf{F}$ and $\mathbf{Q}$ can be leveraged by performing three consecutive matrix-vector multiplications, instead of first forming the full precision matrix and then computing the matrix vector product. This reduces both computations and memory requirements.

To compute the joint mean $\boldsymbol{\mu} = \mathbf{F}^{-1}\mathbf{c}$ without expensive matrix inversion, the components $\boldsymbol{\mu}_k$ have to be computed iteratively as

$$
\boldsymbol{\mu}_k = \mathbf{F}_k \boldsymbol{\mu}_{k-1} + \mathbf{c}_k .
\tag{24}
$$

In contrast, the information vector $\boldsymbol{\eta} = \Omega \boldsymbol{\mu}$ can be expressed compactly as

$$
\boldsymbol{\eta} = \mathbf{F}^T \mathbf{Q} \mathbf{F} \mathbf{F}^{-1} \mathbf{c} = \mathbf{F}^T \mathbf{Q} \mathbf{c},
\tag{25}
$$

which can be computed efficiently using sparse and parallel matrix-vector multiplications on a GPU. We make use of this property in the DGMRF formulation and in the conjugate gradient method.

### A.2 Extension to higher-order Markov processes

We can easily adjust the joint distribution to accommodate higher-order processes with dependencies on multiple past time steps.

For a $p$-th order Markov process, the dynamics are defined by equations

$$
\begin{aligned}
\mathbf{x}_0 &= \boldsymbol{\mu}_0 + \boldsymbol{\epsilon}_0 & \boldsymbol{\epsilon}_0 &\sim \mathcal{N}(\mathbf{0}, \mathbf{Q}_0^{-1}) \\
\mathbf{x}_1 &= \mathbf{F}_{1,1} \mathbf{x}_0 + \mathbf{c}_1 + \boldsymbol{\epsilon}_1 & \boldsymbol{\epsilon}_1 &\sim \mathcal{N}(\mathbf{0}, \mathbf{Q}_1^{-1}) \\
&\cdots \\
\mathbf{x}_k &= \mathbf{F}_{k,1} \mathbf{x}_{k-1} + \mathbf{F}_{k,2} \mathbf{x}_{k-2} + \cdots + \mathbf{F}_{k,p} \mathbf{x}_{k-p} + \mathbf{c}_k + \boldsymbol{\epsilon}_k & \boldsymbol{\epsilon}_k &\sim \mathcal{N}(\mathbf{0}, \mathbf{Q}_k^{-1}) \\
&\cdots \\
\mathbf{x}_K &= \mathbf{F}_{K,1} \mathbf{x}_{K-1} + \mathbf{F}_{K,2} \mathbf{x}_{K-2} + \cdots + \mathbf{F}_{K,p} \mathbf{x}_{K-p} + \mathbf{c}_K + \boldsymbol{\epsilon}_K & \boldsymbol{\epsilon}_K &\sim \mathcal{N}(\mathbf{0}, \mathbf{Q}_K^{-1}).
\end{aligned}
$$

Following the same steps as before, this results in a linear system

$$
\underbrace{
\begin{bmatrix}
\mathbf{I} \\
-\mathbf{F}_{1,1} & \mathbf{I} \\
-\mathbf{F}_{2,2} & -\mathbf{F}_{2,1} & \mathbf{I} \\
\cdots & \cdots & \cdots & \cdots \\
-\mathbf{F}_{p,p} & -\mathbf{F}_{p,p-1} & \cdots & -\mathbf{F}_{p,1} & \mathbf{I} \\
& \cdots & \cdots & \cdots & \cdots & \cdots \\
& & -\mathbf{F}_{K,p} & \cdots & -\mathbf{F}_{K,2} & -\mathbf{F}_{K,1} & \mathbf{I}
\end{bmatrix}}_{=\mathbf{F}}
\cdot
\underbrace{\begin{bmatrix} \mathbf{x}_0 \\ \mathbf{x}_1 \\ \mathbf{x}_2 \\ \vdots \\ \mathbf{x}_K \end{bmatrix}}_{=\mathbf{x}}
=
\underbrace{\begin{bmatrix} \boldsymbol{\mu}_0 \\ \mathbf{c}_1 \\ \mathbf{c}_2 \\ \vdots \\ \mathbf{c}_K \end{bmatrix}}_{=\mathbf{c}}
+
\underbrace{\begin{bmatrix} \boldsymbol{\epsilon}_0 \\ \boldsymbol{\epsilon}_1 \\ \boldsymbol{\epsilon}_2 \\ \vdots \\ \boldsymbol{\epsilon}_K \end{bmatrix}}_{=\boldsymbol{\epsilon}} .
\tag{26}
$$

This means that the matrix $\mathbf{F}$ is extended by adding the higher-order transition matrices $(\mathbf{F}_{\tau,\tau}, \ldots, \mathbf{F}_{K,\tau})$ to the $\tau$-th lower block diagonal of $\mathbf{F}$ for all $\tau = 1, \ldots, p$. The expressions for $\Omega, \boldsymbol{\mu}$ and $\boldsymbol{\eta}$ remain the same (using the extended $\mathbf{F}$), resulting in a block $p$-diagonal precision matrix.

## B  Experimental details

### B.1  Advection-diffusion process

The advection-diffusion dataset is a random sample from a ST-DGMRF for which the transition matrices are defined according to an advection-diffusion process

$$
\frac{\partial \rho(t, s)}{\partial t} = D \nabla^2 \rho(t, s) - \nabla \cdot (\mathbf{v} \rho(t, s))
\tag{27}
$$

with constant diffusion coefficient $D$ and velocity vector $\mathbf{v} = [u, v]^T$. The process is discretized on a $30 \times 30$ lattice with grid cell size $\Delta x = \Delta y = 1$ and periodic boundary conditions. The spatial discretization results in a system of ordinary differential equations

$$\frac{\partial \boldsymbol{\rho}(t)}{\partial t} = \mathbf{M}\boldsymbol{\rho}(t), \tag{28}$$

where $\boldsymbol{\rho}(t)$ is a vector containing the system states of all grid cells. Using a finite difference discretization, matrix $\mathbf{M}$ is defined as

$$\mathbf{M}_{ij} = \begin{cases} D - \frac{1}{2}\mathbf{n}_{ij}^T\mathbf{v} & \text{if } d(i,j) = 1 \\ -4D & \text{if } i = j \\ 0 & \text{otherwise,} \end{cases} \tag{29}$$

where $d(i,j)$ denotes the distance between cell $i$ and $j$, and $\mathbf{n}_{ij}$ denotes the unit vector pointing from lattice cell $i$ to its neighbor $j$. For example, for cell $i = (s_x, s_y)$ and cell $j = (s_x, s_y - 1)$ it is $\mathbf{n}_{ij} = [0, -1]^T$, and thus $\mathbf{n}_{ij}^T\mathbf{v} = -v$.

Eq. 28 is converted into a discrete-time dynamical system by approximating

$$\boldsymbol{\rho}_{t+\Delta t} = \exp(\Delta t \cdot \mathbf{M})\boldsymbol{\rho}_t \approx \left( \sum_{k=0}^{3} \frac{1}{k!}(\Delta t)^k (\mathbf{M})^k \right) \boldsymbol{\rho}_t = \mathbf{F}_{\text{adv-diff}}\boldsymbol{\rho}_t \tag{30}$$

using a third-order Taylor series expansion. For simplicity, we use time resolution $\Delta t = 1$ resulting in

$$\mathbf{F}_{\text{adv-diff}} = \mathbf{I} + \mathbf{M} + \frac{1}{2}\mathbf{M}^2 + \frac{1}{6}\mathbf{M}^3. \tag{31}$$

### B.1.1 Process simulation

We sample the initial state $\boldsymbol{\rho}_0$ from a GMRF with $\boldsymbol{\mu}_0 = \mathbf{0}$ and precision matrix $\mathbf{Q}_0 = \mathbf{S}_0^T\mathbf{S}_0$ with $\mathbf{S}_0 = (\mathbf{D} - \mathbf{A})$, where $\mathbf{A}$ is the adjacency matrix of the 4-nearest neighbour graph $\mathcal{G}_{\text{lattice}}$, and $\mathbf{D} = 4 \cdot \mathbf{I}$ is the corresponding degree matrix. This corresponds to a 1-layer DGMRF with parameters $\alpha = 1, \beta = -1, \gamma = 1$ and $b = 0$.

Starting from $\boldsymbol{\rho}_0$, we iteratively sample the next system state according to

$$\boldsymbol{\rho}_k = \mathbf{F}_k\boldsymbol{\rho}_{k-1} + \boldsymbol{\epsilon}_k \quad \boldsymbol{\epsilon}_t \sim \mathcal{N}(\mathbf{0}, \mathbf{Q}_k^{-1}) \tag{32}$$

with time-invariant transition matrix $\mathbf{F}_k = (\mathbf{F}_{\text{adv-diff}})^4$, where $\mathbf{F}_{\text{adv-diff}}$ is defined according to Eq. (31). This effectively aggregates four simulation steps into one, i.e. $\Delta t_k = (t_{k+1} - t_k) = 4$, resulting in larger differences between consecutive system states. For the noise terms $\boldsymbol{\epsilon}_k$, we use a time-invariant precision matrix $\mathbf{Q}_k = \mathbf{S}_k^T\mathbf{S}_k$ where $\mathbf{S}_k = (10 \cdot \mathbf{I} - \mathbf{A})$. This corresponds to a 1-layer DGMRF with parameters $\alpha = \frac{10}{4}, \beta = -1, \gamma = 1$ and $b = 0$.

We simulate for $K = 20$ time steps, using $D = 0.01$ and $\mathbf{v} = [-0.3, 0.3]^T$, and generate observations by masking out grid cells within a square of width $w \in \{6, \ldots, 12\}$ for 10 consecutive time steps and applying white noise with standard deviation $\sigma = 0.01$.

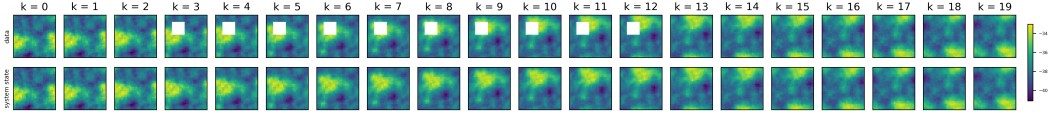

Figure 5: Advection-diffusion dataset with ground truth system states (bottom) and corresponding observations using masks of width $w = 9$ (top).

### B.1.2 ST-DGMRF parameterization

We consider two ST-DGMRF variants that capture different amounts of prior knowledge. In both cases, spatial and temporal layers are defined based on $\mathcal{G}_{\text{lattice}}$, and temporal bias terms are, similar to spatial bias terms, defined as $\mathbf{b}_f^{(l)} = b_f^{(l)}\mathbf{1}$.

**Variant 1**   If prior knowledge is available in the form of a parameterized transition model, the ST-DGMRF transition matrices can be parameterized accordingly. Here, we consider temporal layers $\mathbf{F}_k^{(l)}$ that are a simplified first-order approximation to the true transition matrix $\mathbf{F}_{\text{adv-diff}}$ used to generate the data (see Eq. (31)), i.e. $\mathbf{F}_k^{(l)} = \mathbf{I} + \mathbf{M}^{(l)}$, with time-invariant learnable diffusion coefficients $D^{(l)}$ and velocity vectors $\mathbf{v}^{(l)}$. To ensure that the diffusion coefficient is non-negative, we model it as $D^{(l)} = (d^{(l)})^2$. This leaves us with four learnable parameters $d^{(l)}, u^{(l)}, v^{(l)}$ and $b_f^{(l)}$ per temporal layer.

**Variant 2**   If only partial knowledge about the underlying dynamics is available, the unknown parts can, for example, be replaced by a small neural network. Here, we consider temporal layers of the form $\mathbf{F}_k^{(l)} = \mathbf{I} + \mathbf{M}^{(l)}$ with

$$\mathbf{M}_{ij}^{(l)} = \begin{cases} (d^{(l)})^2 + \phi_{ij,1}^{(l)} & \text{if } j \in n(i) \\ -4(d^{(l)})^2 + \sum_{j \in n(i)} \phi_{ij,2}^{(l)} & \text{if } i = j \\ 0 & \text{otherwise,} \end{cases} \tag{33}$$

where we define $\phi_{ij,1}^{(l)}, \phi_{ij,2}^{(l)} = f_{MLP}^{(l)}(\mathbf{n}_{ij})$ where $f_{MLP}^{(l)} : \mathbb{R}^2 \rightarrow \mathbb{R}^2$ is a multilayer perceptron (MLP) with one hidden layer of width 16 with ReLU non-linearity, and Tanh output non-linearity. Again, we define the transition model to be time-invariant and share MLP parameters across time and space. This amounts to 83 learnable parameters per temporal layer.

**Log-determinant computations**   In our experiments, the spatial base graph $\mathcal{G}_{\text{lattice}}$ is small enough to pre-compute eigenvalues exactly and use the eigenvalue method for log-determinant computations proposed in [43].

**Variational distribution**   For the variational distribution, we also consider two variants, one without temporal dependencies (equivalent to the DGMRF baseline) and one with a single temporal layer with time-invariant *diffusion* transition matrices $\tilde{\mathbf{F}}_k = \lambda \mathbf{I} + \omega(\mathbf{A} - \mathbf{D})$. Note that for $\omega = 0$, this reduces to a simple auto-regressive process.

**Observation model**   All ST-DGMRF variants assume a temporally and spatially invariant observation noise level of $\sigma = 0.01$. The observation matrices $\mathbf{H}_k$ are defined as selection matrices matching the training masks during the learning phase and the training plus validation masks during the testing phase.

### B.2   Air quality data

The air quality dataset is based on hourly PM2.5 measurements obtained from [59]. We consider 246 sensors within the metropolitan area of Beijing, China, for which we extracted time series of $K = 400$ hours between 13 March 2015 at 12pm and 30 March 2015 at 3am. Relevant weather covariates (surface temperature, as well as u and v wind components at 10 meters above ground level) were extracted from the ERA5 reanalysis dataset [25].

#### B.2.1   Data preprocessing

We define both the spatial and the temporal base graph based on the Delaunay triangulation of sensor locations, $\mathcal{G}_{\text{Delaunay}}$, where we disregard edges between sensors that are more than 160 kilometers apart. Edge weights are defined as the inverse distance between sensors, normalized to range between 0 and 1. The raw PM2.5 measurements are log-transformed and standardized to zero mean and unit variance. Finally, we remove clear outliers where the transformed values jump up and down by more than a threshold of $\delta = 2.0$ within three consecutive time steps. The ERA5 covariates are normalized to range between -1 and 1.

To mimic a realistic setting of repeatedly occurring partial network failures, we define our test set by masking out all measurements within a predefined spatial block (containing 50% of all sensors) within 10 randomly placed windows of 20 time steps (see Figure 6). Note that these windows may overlap, resulting in fewer periods of missing data with variable length. The masked out measurements are used for the final model evaluation.

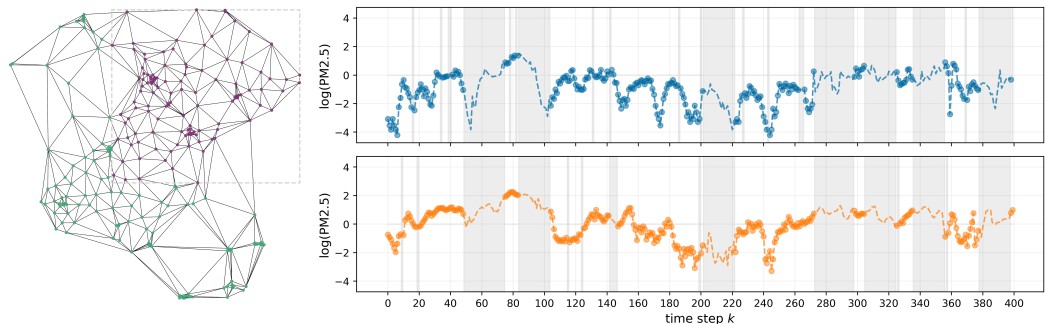

Figure 6: Left: air quality sensor network. Ca. 50% of the nodes are masked out (purple nodes within the gray box) during 10 randomly placed (partially overlapping) windows of 20 time steps. Right: associated log-transformed and normalized PM2.5 measurements for two sensors falling within the masked area. Time points that have either missing data or fall within a masked time window are shaded in gray.

### B.2.2 ST-DGMRF parameterization

As with the advection-diffusion dataset, we consider two ST-DGMRF variants with different types of temporal layers. In both cases, spatial and temporal layers are defined based on the Delaunay triangulation described in Section B.2.1, and temporal bias terms $\mathbf{b}_f$ are defined in terms of a neural network mapping local weather covariates to temporally and spatially varying biases. We use a simple MLP with one hidden layer of width 16 with ReLU activations and no output non-linearity. The MLP parameters are shared over both space and time.

**Variant 1** This variant accounts for directional transport processes, adopting a transition model similar to the neural network model used in the advection-diffusion experiments. In particular, we consider temporal layers of the form $\mathbf{F}_k^{(l)} = \mathbf{I} + \mathbf{M}_k^{(l)}$ with

$$\left(\mathbf{M}_k^{(l)}\right)_{ij} = \begin{cases} (d^{(l)})^2 + \phi_{k,ij}^{(l)} & \text{if } j \in n(i) \\ -4(d^{(l)})^2 + \sum_{j \in n(i)} \psi_{k,ij}^{(l)} & \text{if } i = j \\ 0 & \text{otherwise,} \end{cases} \tag{34}$$

where we define $\phi_{k,ij}^{(l)}, \psi_{k,ij}^{(l)} = f_{MLP}^{(l)}\left(\mathbf{n}_{ij}, w_{ij}, (\mathbf{u}_k)_i\right)$ where $f_{MLP}^{(l)} : \mathbb{R}^6 \to \mathbb{R}^2$ is a MLP with one hidden layer of width 16 with ReLU activations, and Tanh output non-linearity. $w_{ij}$ are the edge weights of the base graph (see Section B.2.1), and $(\mathbf{u}_k)_i$ is the vector of weather covariates for node $i$ at time $k$. Since we use these time-dependent covariates as input to the MLP, the resulting transition model is not time-invariant anymore. However, the parameters of the MLP remain shared across time and space. As before, diffusion parameter $d^{(l)}$ is assumed to be spatially and temporally invariant.

**Variant 2** The second variant uses highly simplified *diffusion* temporal layers of the form $\mathbf{F}_k^{(l)} = \lambda^{(l)}\mathbf{I} + \omega^{(l)}(\mathbf{A} - \mathbf{D})$ with spatially and temporally invariant parameters $\lambda^{(l)}$ and $\omega^{(l)}$.

**Log-determinant computations** Again, the spatial base graph $\mathcal{G}_{\text{Delaunay}}$ is small enough to pre-compute eigenvalues exactly and use the eigenvalue method for log-determinant computations [43].

**Variational distribution** As with the advection-diffusion dataset, we consider two variants for the variational distribution, one without temporal dependencies and one with a single temporal *diffusion* layer.

**Observation model** All ST-DGMRF variants assume a temporally and spatially invariant observation noise level of $\sigma = 0.01$. The observation matrices $\mathbf{H}_k$ are defined as selection matrices matching the training masks during the learning phase and the training plus validation masks during the testing phase.

### B.3 Baseline models

#### B.3.1 DGMRF

We apply the DGMRF for general graphs introduced by [43] to each time frame of the time series, not accounting for temporal dependencies. The DGMRF parameters are not shared across time, allowing for dynamically changing spatial covariance patterns. We use one spatial layer in the variational distribution, as proposed in [43], and run a hyperparameter search over $L_{\text{spatial}} \in \{1, 2, 3\}$ with $L_{\text{spatial}} = 2$ performing best.

**Including covariates** In our experiments on the air quality dataset, for which we have access to relevant covariates, we follow [50] and add linear effects to the measurement model. Note that the vector of coefficients if shared across both space and time.

#### B.3.2 ARMA

We implemented ARMA$(p, q)$ models with $p = 1$ and $q = 1$ for the advection-diffusion data, and with $p = 2$ and $q = 2$ for the air quality data, using the Python `statsmodels` package. For each node in the test set, maximum likelihood parameter estimation is performed based on the observed time points. Given the estimated model coefficients, we obtain posterior mean and variance estimates using the standard Kalman smoother [47]. As the maximum likelihood estimates are deterministic, we do not provide standard deviations of the evaluation metrics for these models.

#### B.3.3 ST-AR

The spatiotemporal autoregressive (ST-AR) model takes the form $\mathbf{x}_k = \alpha \cdot \mathbf{x}_{k-1} + \boldsymbol{\epsilon}_k$, with initial state $\mathbf{x}_0 \sim \mathcal{N}(\boldsymbol{\mu}_0, \boldsymbol{\Sigma}_0)$ and unconstrained spatial error terms $\boldsymbol{\epsilon}_k \sim \mathcal{N}(\mathbf{0}, \mathbf{Q}^{-1})$. We fix $\boldsymbol{\Sigma}_0 = 10 \cdot \mathbf{I}$ to encode high uncertainty about the initial state $\mathbf{x}_0$, and fit $\alpha$, $\boldsymbol{\mu}_0$ and $\mathbf{Q}^{-1}$ to the data using closed-form EM updates. The EM algorithm is initialized with $\alpha = 1$, $\boldsymbol{\mu}_0 = \mathbf{0}$ and $\mathbf{Q}^{-1} = \text{diag}(\mathbf{q})$ where elements $\mathbf{q}_i$ are drawn randomly from the interval $[5, 6]$. After convergence of the EM-algorithm, the final state estimates are obtained with the Kalman smoother [47].

#### B.3.4 EnKS

We consider an Ensemble Kalman Smoother (EnKS) variant for which the transition model matches the true data-generating process of the advection-diffusion dataset, as well as an EnKS variant for which we use a state augmentation approach to estimate unknown parameters $\mathbf{v}$ and $d = \sqrt{D}$ jointly with the system states. For both variants, we use $10^4$ ensemble members (the maximum feasible on our machine). We fix the initial state distribution to $\mathbf{x}_0 \sim \mathcal{N}(\mathbf{0}, 10 \cdot \mathbf{I})$, and sample transition noise terms as $\boldsymbol{\epsilon}_k \sim \mathcal{N}(\mathbf{0}, 0.1 \cdot \mathbf{I})$.

For the state augmentation approach, we define the initial distribution over velocities $\mathbf{v}$ as $\mathcal{N}(\boldsymbol{\mu}_v, 0.1 \cdot \mathbf{I})$, where $\boldsymbol{\mu}_v$ is randomly drawn from $[-1, 1]$ for each repeated run of the EnKS. Similarly, the initial distribution for diffusion parameter $d$ is defined as $\mathcal{N}(\mu_d, 0.01)$ where $\mu_d$ is randomly drawn from $[0, 0.2]$ for each repeated run of the EnKS. Finally, the transition noise terms for parameters $\mathbf{v}$ and $d$ are sampled from $\mathcal{N}(\mathbf{0}, 0.01 \cdot \mathbf{I})$.

#### B.3.5 MLP

For the air quality dataset, the MLP baseline maps local weather covariates $(\mathbf{u}_k)_i \in \mathbb{R}^3$ to log-transformed PM2.5 measurements. We use one hidden layer of width 16 with ReLU activations and no output non-linearity. The MLP parameters are shared over both space and time.

### B.4 Regularized Conjugate Gradients

We use a regularized variant of the conjugate gradient (CG) method [4] to avoid slow convergence in the case of ill-conditioned matrices. Instead of directly solving a potentially ill-conditioned linear system $\mathbf{A}\mathbf{x} = \mathbf{b}$, the idea is to iteratively solve a sequence of regularized (i.e. well conditioned) linear systems

$$(\nu\mathbf{I} + \mathbf{A})\mathbf{x} = \nu\mathbf{x}^{(i)} + \mathbf{b}. \tag{35}$$

At each iteration, the solution from the previous iteration $\mathbf{x}^{(i)}$ is used to obtain the next solution $\mathbf{x}^{(i+1)}$. Eventually, this sequence will converge towards the true solution $\mathbf{x}^*$ of the original system $\mathbf{A}\mathbf{x} = \mathbf{b}$.

We start with $\nu = 10$ and decrease it every 10 iterations by factor 10. In each iteration, the standard CG method is employed to iteratively solve the regularized linear system until the residual norm drops below a threshold of $10^{-7}$ or a maximum of 200 inner CG iterations is reached. This inner loop is repeated until the norm of the residuals

$$\mathbf{r}^{(i)} = \left((\nu\mathbf{I} + \mathbf{A})\mathbf{x}^{(i)}\right) - \left(\nu\mathbf{x}^{(i)} + \mathbf{b}\right) \tag{36}$$

drops below a threshold of $10^{-7}$ or a maximum of 100 outer iterations is reached. The initial guess $\mathbf{x}^{(0)}$ is given by the mean of the variational distribution $q_\phi(\mathbf{x})$.

## C   Additional results

In this section, we present additional results regarding the scalability of our approach (Section C.1), and provide more detailed results for the experiments in Section 4 of the main paper (Section C.2 and C.3). Finally, in Section C.4 we provide estimates of the total computation time required for our experiments.

### C.1   Scalability

To empirically demonstrate the scalability of our method, we generate additional advection-diffusion datasets with varying lattice size and compare the runtime of our ST-DGMRF approach to a naive Kalman smoother (KS) [47] approach. To this end, we consider a model with advection-diffusion transition matrix using $L_{\text{temporal}} = 2$ temporal and $L_{\text{spatial}} = 2$ spatial layers. To avoid additional matrix inversions in the KS approach, we set the spatial and temporal bias terms $\mathbf{b}_s, \mathbf{b}_f$ to zero, resulting in $\boldsymbol{\mu}_0 = \mathbf{0}$ and $\mathbf{c}_k = \mathbf{0}$. We train the model for $1\,000$ iterations and measure the average wall clock time per iteration. In addition, we measure the wall clock time needed to perform inference with the trained model.

**ST-DGMRF**   For the ST-DGMRF approach, we proceed as before using a variational distribution with one temporal diffusion layer during training. For better comparability, we employed the standard (non-regularized) CG method (with a tolerance of $10^{-7}$) to compute the posterior mean and marginal variances (based on 100 CG samples) and provide measurements of the average time per CG iteration instead of the total time needed to perform inference. Multiplying this with the average number of CG iterations needed until convergence results in an estimate of the average total time for inference.

**Kalman smoother**   For the KS approach, instead of approximating the true posterior with a variational distribution and estimating the ELBO based on Monte-Carlo samples, we use the KS to obtain exact marginal posterior estimates, which are used to compute expectations in closed form. The marginal covariance and transition matrices required for the KS equations are extracted from the ST-DGMRF model in every iteration. The associated parameters are then optimized via a form of Generalized EM-algorithm [16], where in each iteration a single gradient ascent step is taken.

**Results**   Figure 7 shows how ST-DGMRF and KS training and inference scale as the number of nodes $N$ in the system increases. Clearly, the time per training iteration increases super-linearly when the KS is used to obtain exact marginal posterior distributions, while the variational ST-DGMRF training time increases only marginally and remains below the fastest KS iteration for all tested $N$. In addition, KS memory requirements (due to storing $K$ dense $N \times N$ covariance matrices) exceeded the available GPU memory for $N > 1024$, making this approach infeasible for larger systems. In contrast, the ST-DGMRF exploits the sparsity of spatial and temporal graph-structured layers and thereby avoids storing dense matrices, remaining feasible for $N \gg 1024$.

For the tested systems, ST-DGMRF posterior inference with the CG method is slower than exact KS inference. However, the memory requirements of the KS approach again limit its feasibility to $N \leq 1024$, while the CG method only requires storing vectors of size $\mathcal{O}(N)$ making it feasible for $N \gg 1024$. Moreover, Figure 7 confirms that computations per CG iteration scale linearly in $N$. And since the number of CG iterations required for convergence remains approximately constant, the

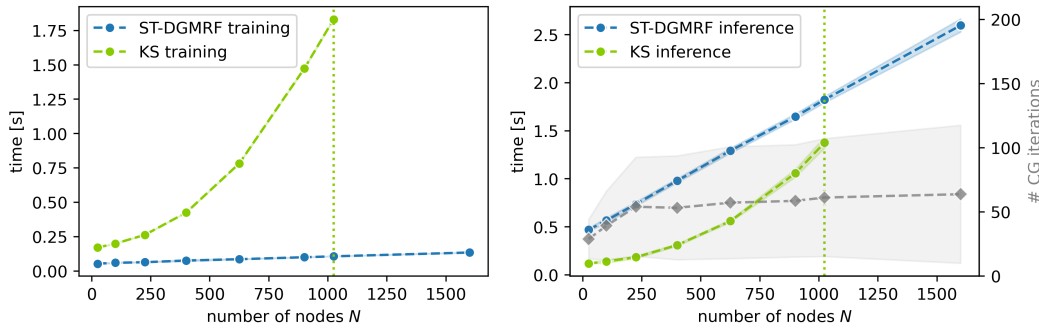

Figure 7: Comparison of ST-DGMRF and KS computation time in seconds for training (per iteration) and inference. For ST-DGMRF inference, the time per CG iteration is plotted together with the average number of CG iterations needed to converge (in gray). All quantities are plotted as mean $\pm$ std based on 5 runs with different random seeds. The vertical dotted lines indicate the maximum $N$ for which the KS approach was applicable.

total computation time for CG inference also scales linearly in $N$. In contrast, KS inference again scales super-linearly. This means that even if the KS approach would remain feasible in terms of memory requirements, its computation time will quickly approach, and eventually exceed, the time needed for CG inference.

## C.2 Advection-diffusion experiments

Table 4 summarizes all results for the advection-diffusion dataset with mask size $w = 9$, including standard deviations for all metrics based on 5 runs with different random seeds. As discussed in the main paper, the ST-DGMRF variants provide more accurate posterior estimates than the baselines relying on simplified spatiotemporal dependency structures.

**Ablation results** Table 4 contains additional results for the ST-DGMRF variants using different settings for the variational distribution (see Section B.1.2) For this dataset, we do not find a significant effect of introducing temporal dependencies in the variational distribution. Further, Figure 8 shows additional results for the ST-DGMRF variants when varying the number of temporal layers $L_{\text{temporal}}$. For all metrics, the performance improves significantly as we start adding temporal layers and stabilizes around $L_{\text{temporal}} = 3$. Note that around the same point, both ST-DGMRF variants converge towards the EnKS using the true data-generating dynamics, in terms of the $\text{RMSE}_\mu$, and even drop below it in terms of the CRPS. Only in terms of $\text{RMSE}_\sigma$, the ST-DGMRF models remain inferior to both EnKS variants. We hypothesize that increasing the expressivity (i.e. $L_{\text{spatial}}$) of the noise terms can further reduce this gap.

Table 4: Model performance for the advection-diffusion dataset with $w = 9$, reported as mean $\pm$ std over 5 runs with different random seeds. All ST-DGMRF variants use $L_{\text{spatial}} = 2$ and $L_{\text{temporal}} = 4$.

| | VI dynamics | $\text{RMSE}_\mu \downarrow$ | $\text{RMSE}_\sigma \downarrow$ | CRPS $\downarrow$ |
|---|---|---|---|---|
| ARMA | – | 2.3054 – | 0.6812 – | 1.7064 – |
| ST-AR | – | $1.4595_{\pm 0.0098}$ | $1.9216_{\pm 1.0392}$ | $0.9707_{\pm 0.0163}$ |
| DGMRF | – | $0.5901_{\pm 0.0037}$ | $0.3808_{\pm 0.0010}$ | $0.3495_{\pm 0.0022}$ |
| EnKS | | | | |
| *true dynamics* | – | $0.0661_{\pm 0.0030}$ | $0.0046_{\pm 0.0000}$ | $0.1027_{\pm 0.0035}$ |
| *estimated dynamics* | – | $0.1654_{\pm 0.2031}$ | $0.0039_{\pm 0.0005}$ | $0.1434_{\pm 0.0902}$ |
| ST-DGMRF (ours) | | | | |
| *advection-diffusion* | *none* | $0.0526_{\pm 0.0001}$ | $0.1148_{\pm 0.0003}$ | $0.0726_{\pm 0.0001}$ |
| *advection-diffusion* | *diffusion* | $0.0526_{\pm 0.0001}$ | $0.1146_{\pm 0.0005}$ | $0.0726_{\pm 0.0000}$ |
| *neural network* | *none* | $0.0839_{\pm 0.0022}$ | $0.1334_{\pm 0.0089}$ | $0.0833_{\pm 0.0008}$ |
| *neural network* | *diffusion* | $0.0854_{\pm 0.0027}$ | $0.1402_{\pm 0.0061}$ | $0.0839_{\pm 0.0008}$ |

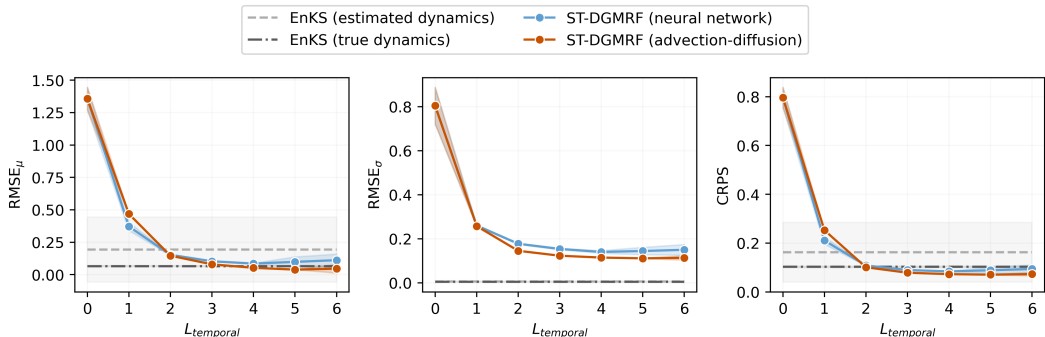

Figure 8: $\text{RMSE}_\mu$, $\text{RMSE}_\sigma$ and CRPS as a function of the number of temporal layers $L_{\text{temporal}}$ for the advection-diffusion dataset with $w = 9$, plotted as mean $\pm$ std over 5 runs with different random seeds. Both ST-DGMRF variants are trained with a variational distribution using one temporal *diffusion* layer. Note that $L_{\text{temporal}} = 0$ corresponds to the spatial-only DGMRF baseline.

## C.3 Air quality experiments

Table 5 summarizes all results for the air quality dataset. It contains additional results for the ST-DGMRF variants using different settings for the variational distribution (see Section B.2.2), and provides standard deviations for all metrics based on 5 runs with different random seeds.

**Ablation results** We find that, in contrast to our experiments on the advection-diffusion data, accounting for temporal dependencies in the variational distribution is clearly beneficial in the real world setting. Especially for the ST-DGMRF with neural network based transitions, adding the temporal *diffusion* layer results in significantly improved posterior estimates, and at the same time reduces the variability across different runs. Further, we find that at least two temporal layers are needed to achieve good posterior estimates that improve on the baselines (see Figure 9).

**Example model outputs** Figure 10 shows state estimates and associated uncertainties together with sensor measurements for two example sensors within the masked out area of the network, for ST-DGMRF, DGMRF and ARMA respectively. For all three models, state estimates are obtained by conditioning on the input data points (used for training), resulting in low errors and uncertainties for observed time points and higher errors and uncertainties for masked out time points. Moreover, for both ST-DGMRF and DGMRF, higher uncertainties coincide with larger errors and larger fluctuations in the measurements (top), while more accurate state estimates come with smaller uncertainties (bottom). Finally, Figure 11 visualizes how spatial and temporal ST-DGMRF layers transform samples from the estimated posterior over system states into (approximately) independent Gaussian noise, as derived in Section 3.1.2 and visualized in Figure 1. Clearly, temporal layers remove daily patterns and overall trends, while spatial layers remove dependencies between close-by sensors and increase temporal fluctuations.

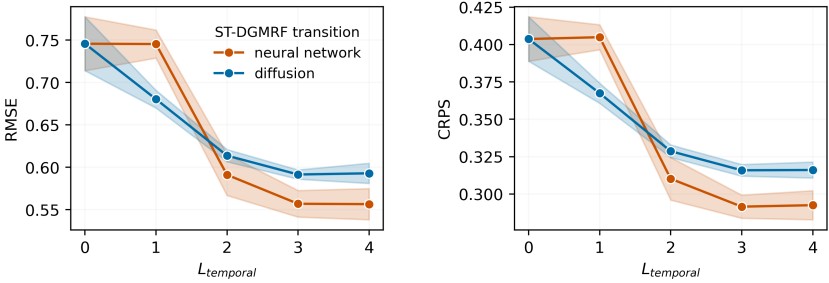

Figure 9: RMSE and CRPS for increasing $L_{\text{temporal}}$ (mean $\pm$ std over 5 runs). Both models use $p = 2$. As before, $L_{\text{temporal}} = 0$ corresponds to the spatial-only DGMRF baseline.

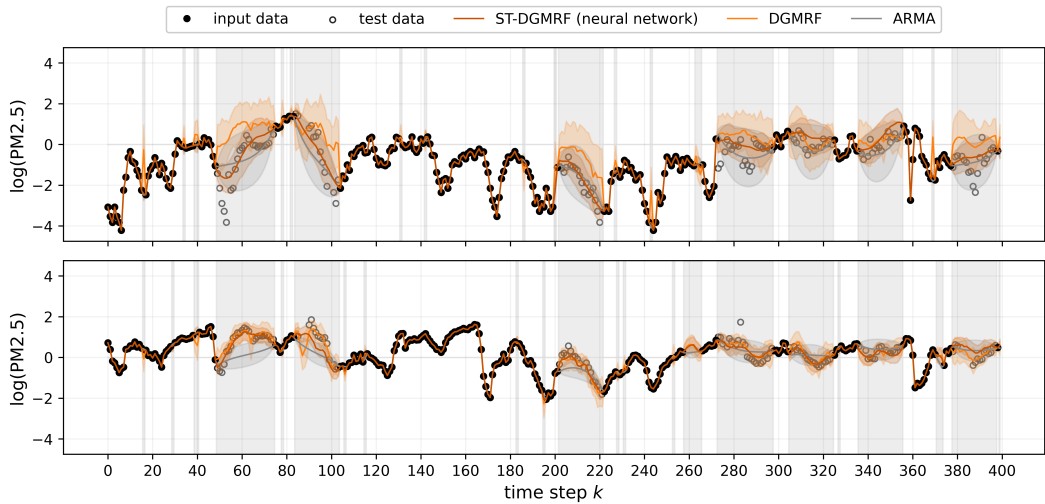

Figure 10: Model outputs for two air quality sensors falling within the masked area. Solid lines represent posterior mean estimates, while shaded areas represent posterior std estimates. Time points that have either missing data or fall within the masked time window are shaded in gray.

Table 5: Model performance for the air quality dataset, reported as mean $\pm$ std over 5 runs with different random seeds. All ST-DGMRF variants use $L_{\text{spatial}} = 2$ and $L_{\text{temporal}} = 4$.

| | $p$ | VI dynamics | RMSE $\downarrow$ | CRPS $\downarrow$ |
|---|---|---|---|---|
| ARMA | – | – | $0.6820$ – | $0.3625$ – |
| ST-AR | – | – | $0.7350_{\pm 0.0006}$ | $0.4261_{\pm 0.0003}$ |
| DGMRF | – | – | $0.7368_{\pm 0.0135}$ | $0.3966_{\pm 0.0032}$ |
| MLP | – | – | $0.8038_{\pm 0.0245}$ | – |
| ST-DGMRF (ours) | | | | |
| *diffusion* | 1 | *none* | $0.6147_{\pm 0.0082}$ | $0.3239_{\pm 0.0058}$ |
| *diffusion* | 1 | *diffusion* | $0.6190_{\pm 0.0073}$ | $0.3258_{\pm 0.0043}$ |
| *diffusion* | 2 | *none* | $0.6020_{\pm 0.0112}$ | $0.3214_{\pm 0.0051}$ |
| *diffusion* | 2 | *diffusion* | $0.5928_{\pm 0.0119}$ | $0.3161_{\pm 0.0054}$ |
| *neural network* | 1 | *none* | $0.5995_{\pm 0.0887}$ | $0.3147_{\pm 0.0494}$ |
| *neural network* | 1 | *diffusion* | $0.5853_{\pm 0.0457}$ | $0.3092_{\pm 0.0257}$ |
| *neural network* | 2 | *none* | $0.5825_{\pm 0.0626}$ | $0.3062_{\pm 0.0353}$ |
| *neural network* | 2 | *diffusion* | $0.5565_{\pm 0.0184}$ | $0.2925_{\pm 0.0097}$ |

## C.4 Total compute

Most computations were performed on a Nvidia Titan X GPU. On top of the final experiments, we performed hyperparameter sweeps and additional test runs. Here, we provide estimates of the total compute time grouped by experiment:

**Advection-diffusion dataset:**

- **Performance comparison & ablations:** ca. 50 GPU hours
- **Varying mask size:** ca. 30 GPU hours per $w$, resulting in ca. 210 GPU hours in total
- **Scalability:** ca. 15 GPU hours

**Air quality dataset:**

- **Performance comparison & ablations:** ca. 100 GPU hours

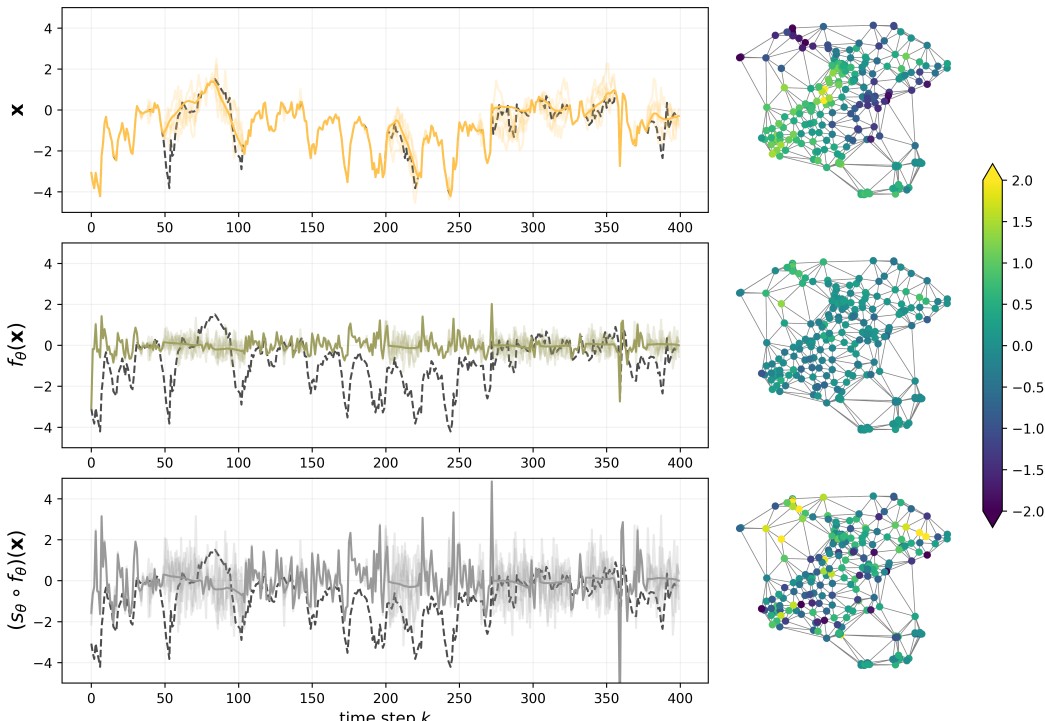

Figure 11: Effects of applying temporal ($\mathbf{f}_\theta$) and spatial ($\mathbf{s}_\theta$) ST-DGMRF layers to the predictive posterior. Left column: The dark yellow line (top row) shows the estimated posterior mean for an air quality sensor falling within the masked area. Light yellow lines represent corresponding posterior samples. Similarly, green lines (center row) represent states after applying the temporal transformation ($\mathbf{f}_\theta$), and gray lines (bottom row) represent states after applying both temporal and spatial layers ($\mathbf{s}_\theta \circ \mathbf{f}_\theta$). As a reference, we also plot ground truth log-transformed and normalized PM2.5 measurements (dashed black lines). Right column: corresponding (transformed) states for all sensors at time $k = 100$.

