# OpenReview forum: "Deep Gaussian Markov Random Fields for Graph-Structured Dynamical Systems"
_NeurIPS.cc/2023/Conference — NeurIPS 2023 poster_

### Official Review · Reviewer_Upf6 · 2023-07-02

**Soundness:** 3 good
**Presentation:** 3 good
**Contribution:** 3 good
**Rating:** 6
**Confidence:** 4

**Summary:**

This paper proposes a Gaussian model for graph-temporal data, with two sorts of sparseness in the precision matrix: blockwise due to an assumed Markov property in time, and within blocks due to assumed sparse graph structures. To achieve inference and parameter learning the paper proposes a two-stage approach with parameter learning done via VI and final posterior (mean) inference with CG. The variational distribution has a suitable structure that reflects the temporal sparseness of the model.




**Strengths:**

The presented model is a very natural and ubiquitous one, so the setting is well motivated. There are no other approaches that I am aware of that solve this particular situation. The approach is certainly valid.

I found this paper easy to read and well explained.

Rather than solve the inference problem with a Kalman-style algorithm (as in [1]) this paper proposes a two-stage approach of using VI for parameter learning and then CG for the final posterior (mean). I initially thought this was a rather strange thing to do, but on reflection I see that it is quite a good idea in this setting. I wonder whether this has more general application.

The design of the posterior distribution is well chosen to capture the (exact?) posterior while reducing the computation complexity of the naive approach.

**Weaknesses:**

The experiments section is rather weak and is comes across more as a unit test than a proper evaluation: the data is drawn from exactly the correct model class, and the primary comparison is to a model that is the same but missing the temporal dependencies. There is not comparison on real data, or on simulated data with any other approach except extremely simple AR/MA/ARMA models. The approach of [1] is applicable for the temporal part of this setting, and for a 30x30 grid it is feasible to take the 900 dimensional dense representation. This would have made a useful comparison.

In terms of novelty, the extension the spatio-temporal regime is not a huge leap, and is already arguably already in the DGMRF model class already. Most of VI is introduced in [40], so the novelty there is not that great.

While the setting is general, to enable fast determinant the form of the spatial layer is actually quite restricted. For a general spatial graph the approach would be no better than kalman approaches.

**Questions:**

With the extension propose on 267 it would seem to me that the variational distribution contains the exact posterior. Is this correct? If so, then the CG could surely be omitted if the VI was run to convergence (e.g. with an annealing learning rate).

I assume the variational distribution is learned by direct gradient descent on the parameters. This would be rather inefficient if the matrices are ill conditioned. Does the natural gradient approach of [1] work here?

The motivation for the two-stage approach is that "parameter estimation becomes computationally prohibitive in high dimensional settings". [57] shows that doing the (truncated) CG algorithm also gives the marginal likelihood log det term with a good approximation. I feel that this is relevant work which should be discussed.

[57] Scalable Log Determinants for Gaussian Process Kernel Learning, Dong, Eriksson, Nickisch, Bindel, Wilson, NeurIPS 2017

Typo: 371 extend -> extent

**Limitations:**

The approach is limited to Gaussian models, and in this case exact inference is available if the problem is small enough, so the method is only relevant for larger data. It is not really made clear in the experiments how large is feasible, and whether the two stage approach really works on real rather than simulated data.

---

> ### Author Rebuttal · Authors · 2023-08-10
>
> # Official Rebuttal to Upf6
>
> We would like to thank Reviewer Upf6 for the positive assessment, the thorough feedback and interesting questions. In the following, we will respond to each point separately, and refer to our general response to all reviewers whenever there is overlap with other reviewers.
>
> ## Responses to Weaknesses
>
> > The experiments section is rather weak and is comes across more as a unit test than a proper evaluation: the data is drawn from exactly the correct model class
>
> We have added additional experiments on a real-world air quality monitoring dataset, showing that our approach generalizes well to more complex real world settings. For more details, please see the general response to all reviewers.
>
> > and the primary comparison is to a model that is the same but missing the temporal dependencies. There is not comparison on real data, or on simulated data with any other approach except extremely simple AR/MA/ARMA models. The approach of [1] is applicable for the temporal part of this setting, and for a 30x30 grid it is feasible to take the 900 dimensional dense representation. This would have made a useful comparison.
>
> We have added an AR model with spatial noise (as suggested by reviewer LAkB) as an additional baseline, taking both spatial and temporal dependencies into account. Fore more details, please see the general response to all reviewers.
>
> While we agree that the approach of [1] could serve as an interesting baseline, we are not sure how feasible it is in practice. To the best of our knowledge there is no code publically available for this approach, and given the remaining time frame we do not think it is doable to implement this ourselves.
> However, we would happily include this baseline if the reviewer could point us to a working implementation.
>
> > While the setting is general, to enable fast determinant the form of the spatial layer is actually quite restricted. For a general spatial graph the approach would be no better than kalman approaches.
>
> It is true that for general dense base graphs $G_{\text{spatial}}$ and $G_{\text{spatial}}$, there is no increase in computational efficiency. However, the idea is that in such a setting the dense graph is likely the result of a chain of more local (and thus sparse) interactions, and the dense adjacency matrix $\mathbf{A}$ can thus be approximated by a composition of sparse matrices.
> For further discussion and experimental validation of the spatial DGMRF approach for general graphs, we would like to point the reviewer to [40].
>
> ## Responses to Questions
>
> > With the extension propose on 267 it would seem to me that the variational distribution contains the exact posterior. Is this correct? If so, then the CG could surely be omitted if the VI was run to convergence (e.g. with an annealing learning rate).
>
> Although this seems to be the case at first sight, the variational distribution is defined in terms of a sparse and factorized covariance matrix, while the true posterior has a sparse precision matrix $\mathbf{\Omega}^+=\mathbf{F}^T\mathbf{S}^T\mathbf{SF} + \mathbf{H}^T\mathbf{R}^{-1}\mathbf{H}$. The corresponding covariance matrix $\mathbf{\Sigma}^+=(\mathbf{\Omega}^+)^{-1}$ will generally be dense. To facilitate fast sampling from the variational distribution, the idea is to approximate the true dense covariance matrix $\mathbf{\Sigma}^+$ with a sparse and factorized covariance matrix $\mathbf{\Lambda}$.
>
> > I assume the variational distribution is learned by direct gradient descent on the parameters. This would be rather inefficient if the matrices are ill conditioned. Does the natural gradient approach of [1] work here?
>
> Thank you for this good suggestion. While we did not run into convergence issues with our experiments, it is definitely worth considering this in the future.
>
> > The motivation for the two-stage approach is that "parameter estimation becomes computationally prohibitive in high dimensional settings". [57] shows that doing the (truncated) CG algorithm also gives the marginal likelihood log det term with a good approximation. I feel that this is relevant work which should be discussed.
>
> The advantage of using CG only to compute the final posterior distribution is that the associated iterations need to be run only once. In contrast, using CG to approximate log-determinants, would require running CG repeatedly until convergence during the training loop. Although CG usually converges fast, we would expect that for large systems this would still result in a major bottleneck.
>
> ## Responses to Limitations
>
> > The approach is limited to Gaussian models, and in this case exact inference is available if the problem is small enough, so the method is only relevant for larger data. It is not really made clear in the experiments how large is feasible,
>
> Figure 3 in the appendix shows the general scaling behaviour of our method, supporting our theoretical analysis in the main paper. Of course, the scalability to even larger systems also depends on additional factors such as hardware constraints. To address this concern, we are planning to explore the limits or our approach during the discussion period, and report our findings as soon as possible.
>
> > and whether the two stage approach really works on real rather than simulated data.
>
> Our additional experiments on the real-world air quality dataset show that our method works well also in more challenging settings with unknown and complex dynamics (see general response to all reviewers for more details).
>
> ## Conclusion
>
> We thank reviewer Upf6 again for the time and effort spent on this review and for posing interesting and thought-provoking questions. Similar to the other reviewers, concerns were mainly raised about the experimental evaluation. We addressed this by adding experiments on a real-world dataset and by including a spatiotemporal ST-AR baseline model. If there are any other changes you would like to see, we are more than happy to discuss them with you.

---

> > ### Comment · Reviewer_Upf6 · 2023-08-17
> > **Comment**
> >
> > I thank the authors for the extensive and detailed reply. My questions were addressed satisfactorily.
> >
> > An implementation of [1] is at https://github.com/secondmind-labs/markovflow, thought it may be quite difficult to use.
> >
> > I continue to be broadly in favour of the paper.

---

### Official Review · Reviewer_k13T · 2023-07-04

**Soundness:** 3 good
**Presentation:** 3 good
**Contribution:** 3 good
**Rating:** 4
**Confidence:** 2

**Summary:**

The paper extends Gaussian Random Fields with precision-based spatiotemporal structure.

**Strengths:**

Clarity: The paper is professionally written, with flawless math. The method is technical and dense, and there are many parts to it. It would have been useful to include illustrations, perhaps in the supplements.

Originality: The paper has moderate novelty. The idea of using spatial and temporal precisions to model random fields is very apt, but these ideas are also well known in general.

Quality: The method is principled and well derived.

**Weaknesses:**

The experimental evaluation is insufficient. There is only a single simple advection example, which is effectively a toy case. There are no real world experiments. Unfortunately a paper like this really needs a real world’y experiment to show that the method has transferable performance. The paper is really good, but I’m leaning on rejection for this reason alone.

The performance is good, but incremental. I’m happy with the neural network variant performance, but there is something wrong if the advection model (that matches the true system perfectly!) can’t fit the simple dynamics, nor improve over neural network. I wonder if this is a sampling or data scarcity issue.

The results are a black box. The purpose of this work is to learn temporal causalities, and spatial couplings. Yet, neither are shown! The paper needs to illustrate what kind of structures the system has learnt, and show if they are accurate or useful, and demonstrate transferable insights.

**Questions:**

None

**Limitations:**

No issues

---

> ### Author Rebuttal · Authors · 2023-08-09
>
> # Official Rebuttal to k13T
>
> We would like to thank Reviewer LAkB for praising the quality and clarity of our method and for providing constructive feedback. In the following, we will respond to each point separately, and refer to our general response to all reviewers whenever there is overlap with other reviewers.
>
> ## Responses to Weaknesses
>
> > The experimental evaluation is insufficient. There is only a single simple advection example, which is effectively a toy case. There are no real world experiments. Unfortunately a paper like this really needs a real world’y experiment to show that the method has transferable performance. The paper is really good, but I’m leaning on rejection for this reason alone.
>
> We have added additional experiments on a real-world air quality monitoring dataset, showing that our approach generalizes well to more complex real world settings. For more details, please see the general response to all reviewers.
>
> > The performance is good, but incremental. I’m happy with the neural network variant performance, but there is something wrong if the advection model (that matches the true system perfectly!) can’t fit the simple dynamics, nor improve over neural network. I wonder if this is a sampling or data scarcity issue.
>
> We are happy to report that we resolved this issue. Our updated results (see Table 1 in the provided PDF) now show a significant improvement of the ST-DGMRF models over the baselines, with the advection model with $L_{\text{temporal}}=4$ temporal layers (matching the true transition model) clearly performing best.
> Nevertheless, please note that the issue was not as severe as it seems: in the original version the table showed results for $L_{\text{temporal}}=1$, which presents a simplification of the true transition model not accounting for longer range dependencies.
>
> > The results are a black box. The purpose of this work is to learn temporal causalities, and spatial couplings. Yet, neither are shown! The paper needs to illustrate what kind of structures the system has learnt, and show if they are accurate or useful, and demonstrate transferable insights.
>
> Please note that we are not claiming to learn causal structures, but rather aim at exploiting prior knowledge about such structures in order to improve inferences about unobserved system states. While in general, learning the true (causal) structure of the system will definitely aid in obtaining better state estimates, in practice it may be sufficient to learn a "good enough" approximation. However, we agree with the reviewer that an evaluation of the learned transition and precision matrices is interesting and could provide useful insights into both the model and the system itself.
> Unfortunately, we haven't had the time to do this, but we are hoping to do so during the discussion period.
>
> ## Conclusion
>
> We thank reviewer k13T again for taking the time to provide thoughtful and constructive feedback, which has helped us to improve our experimental evaluation significantly. We hope that we have resolved your major concerns with regard to the experimental evaluation by adding additional experiments on a real-world dataset and by improving the results on the simulated dataset such that they now align well with the expected outcome. If these changes are to your satisfaction, we kindly ask you to consider revising your initial rating accordingly. We are more than happy to discuss possible solutions to any remaining issues with you.

---

> > ### Comment · Reviewer_k13T · 2023-08-11
> > **resp**
> >
> > Thanks for the response. Including a new experiment is beneficial for the paper, but I am worried that this is too much changes to the paper during review period. Similarly I would prefer the method and results to not change during this time either: the paper should have converged before submission. Finally, my concerns about learned structured insights remain.
> >
> > For these reasons I still vote for rejection, although I am raising my score to 4.

---

> > > ### Author Response · Authors · 2023-08-11
> > > **Response to k13T**
> > >
> > > Thank you for taking the time to carefully consider our changes and engage in a discussion with us.
> > >
> > > First, we would like to point out that while the results (Table 2 in original paper, Table 1 in provided PDF) did change, our proposed method as described in the submitted paper did not. The improved numbers are merely a result of improving the actual implementation (addressing issues like numerical instabilities).
> > >
> > > Second, we have included the real-world experiment and the additional spatiotemporal AR baseline during the rebuttal phase to directly address concerns raised by the reviewers. We would like to point out that Neurips facilitates uploading an extra PDF page with figures and tables for this exact purpose. This, in our opinion, indicates that additions and changes that improve the submitted paper are clearly encouraged.
> > > In any case, we appreciate that our changes were acknowledged as beneficial for the paper in general.
> > >
> > > Regarding the structural insights that can be gained using our method, we will come back to you soon to provide details on how we are planning to evaluate the learned structures.  We would be glad to engage in further discussion with you about this.

---

> > > > ### Author Response · Authors · 2023-08-18
> > > > **Evaluation of learned dynamics**
> > > >
> > > > To address remaining concerns regarding the learned structures, we have now compared the learned transitions for different ST-DGMRF models trained on the advection-diffusion dataset with the true data-generating process, which we have access to in this case. In particular, we extracted the stencil for an arbitrary point on the lattice (all points are equivalent) from the true transition matrix as well as from the learned ones. We then compared them qualitatively by visualizing the local graph structure and the corresponding weights of the stencil. Finally, we computed the Pearson correlation coefficient and the MAE between learned and ground truth weights.
> > > >
> > > > We find that for all ST-DGMRF variants the learned transition model is very close to the ground truth. The more temporal layers are used, the better the agreement. Moreover, even with only 2 temporal layers we learn an adequate approximation of the true transition, with Pearson correlation coefficients remaining above 0.87 (p << 0.01) and MAE remaining below 0.025 (ground truth weights range between -0.26 and 0.36), for both the advection-diffusion and the neural network transition models respectively. This is in good agreement with the corresponding posterior evaluation (Table 1 in the provided PDF).
> > > >
> > > > We are planning to incorporate these results in the main text of the paper.
> > > >
> > > > (also replied to 9F96)

---

### Official Review · Reviewer_LAkB · 2023-07-06

**Soundness:** 3 good
**Presentation:** 2 fair
**Contribution:** 2 fair
**Rating:** 6
**Confidence:** 4

**Summary:**

The author extend work on deep Gaussian Markov random field to a spatio-temporal setting. Earlier it has been explored in spatial and graphical setting. The authors show how to retain the sparsity structure of the previous work, which is  need to be computationally feasible.
They also show the various types of temporal structure can be built in this framework.

**Strengths:**

The extension to a spatio-temporal models is of course very important as many datasets have both a spatial and temporal structure.
Also that the methods are computationally feasible is paramount as the complexity spatio-temporal models tend to be very large.
Each part of the paper is clear written.

**Weaknesses:**

The main weakness for me is the validation of the method only a single simulated data set is rather weak.
If I understand it correctly you are comparing your method to either a pure time series model (AR, MA, ARMA) or a pure spatial (DGMRF?) this does not seems like a fair comparison.
Something simple like a AR processes with spatial noise is standard in statistics.


**Questions:**

In Figure 2 to the right should w be from 6-10 in the it says 6-9,12?
Also how come mae is decreasing when w is increasing?
Why is the average value of the simulations very negative in appendix in the colorbar the value ranges from -35 to -40.
Have you added a mean in the other models, so they can compensate for this?
You should also be able to get the marginal posterior distributions for your model how does they look for the simulations?

**Limitations:**

.

---

> ### Author Rebuttal · Authors · 2023-08-09
>
> # Official Rebuttal to LAkB
>
> We would like to thank Reviewer LAkB for the positive assessment and good suggestions. In the following, we will respond to each point separately, and refer to our general response to all reviewers whenever there is overlap with other reviewers.
>
> ## Responses to Weaknesses
>
> > The main weakness for me is the validation of the method only a single simulated data set is rather weak.
>
> We have added additional experiments on a real-world air quality monitoring dataset, showing that our approach generalizes well to more complex real world settings. For more details, please see the general response to all reviewers.
>
> > If I understand it correctly you are comparing your method to either a pure time series model (AR, MA, ARMA) or a pure spatial (DGMRF?) this does not seems like a fair comparison. Something simple like a AR processes with spatial noise is standard in statistics.
>
> We have added such a process as an additional baseline. For more details, please see the general response to all reviewers.
>
> ## Responses to Questions
>
> > In Figure 2 to the right should w be from 6-10 in the it says 6-9,12?
>
> Well spotted! We have updated the text with the correct values w=6-10.
>
> > Also how come mae is decreasing when w is increasing?
>
> As mentioned in line 346-347, this is an artifact of the different set of pixels used for evaluation. E.g. if the smallest mask exactly covers the "mode" of the advection-diffusion process (yellow pixels in Figure 2 (left)) and a model estimates all pixels to have some low "base state" (blue pixels), the average error between true state and model estimate will decrease as the mask size increases and starts covering more of the surrounding pixels that are close to the "base state".
>
> > Why is the average value of the simulations very negative in appendix in the colorbar the value ranges from -35 to -40. Have you added a mean in the other models, so they can compensate for this?
>
> The simulated data is generated by drawing a sample from the initial state distribution and then simulating forward in time. Even though this initial distribution has zero mean, the initial sample happens to be drawn from the "negative side" of the distribution. We indeed made sure that the overall mean is subtracted before fitting the baseline models.
>
> > You should also be able to get the marginal posterior distributions for your model how does they look for the simulations?
>
> We indeed have access to the true marginal posterior of the simulations. As explained in the general response to all reviewers, we have extended our evaluation to include a direct comparison between our marginal posterior estimates and these true marginal posterior distributions.
>
> ## Conclusion
>
> We thank reviewer LAkB again for taking the time to provide thoughtful feedback and for suggesting a feasible spatiotemporal baseline. We hope that we answered your questions to your satisfaction and we could resolve your concerns w.r.t. the evaluation of our method by adding experiments on a real-world dataset. If there are any other changes you would like to see, we are more than happy to discuss them with you.

---

> > ### Comment · Reviewer_LAkB · 2023-08-12
> >
> > Thank you for your response.
> > You have answered my questions.

---

### Official Review · Reviewer_9F96 · 2023-07-24

**Soundness:** 3 good
**Presentation:** 3 good
**Contribution:** 4 excellent
**Rating:** 6
**Confidence:** 4

**Summary:**

This work extends DGMRF to account for temporal dependency in data with a SSM reformulation. With necessary assumptions, the proposed method produces accurate posterior, competitive performance to DGMRF and faster inference to Kalman smoother.

**Strengths:**

- The method extends DGMRF to account for temporal dependency in data.
- It has been shown that the method has low computational complexity and good scalability.
- The method produces higher accurate posterior comparing to DGMRF and other transitional TS models.

**Weaknesses:**

- Since the method is extending to dynamical systems, DGMRF is a bit of "strawman". The method should be also compared with other filter/smoother e.g. deep KL, ensemble KL, particle smoothing and etc.
- Though the posterior was evaluated in the manuscript, the dynamical model (F) wasn't. It is very often that you got a decent posterior but learned a "bad" F (in terms of forecasting).
- Only the posterior mean was evaluated. It is unclear how good the posterior variance is.
- The variational posterior should be compared to the true posterior in an example where the latter is accessible analytically or via sampling.
- Only one synthetic example is not sufficient for evaluating the method thoroughly. It is unclear how it generalizes.

**Questions:**

- Were the observation parameters $H$ and $\xi$ trainable or fixed at the true values?
- Were the hyperparameters such as state noise and observation noise trainable or fixed at the true values?

**Limitations:**

The technical limitations were discussed in the last section.

---

> ### Author Rebuttal · Authors · 2023-08-09
>
> # Official Rebuttal to 9F96
>
> We would like to thank Reviewer 9F96 for the positive feedback and the good suggestions. In the following, we will respond to each point separately, and refer to our general response to all reviewers whenever there is overlap with other reviewers.
>
> ## Responses to Weaknesses
>
> > Since the method is extending to dynamical systems, DGMRF is a bit of "strawman". The method should be also compared with other filter/smoother e.g. deep KL, ensemble KL, particle smoothing and etc.
>
> We added a spatiotemporal ST-AR baseline (see general rebuttal) to address this limitation. Unfortunately, other more complex filter/smoother approaches are more difficult to apply in the setting we are considering. For example, deep Kalman filtering with KVAE [17] requires the definition of a suitable encoder mapping from high-dimensional observations to a latent space. In settings where the pattern of missing observations varies over time, this requires some initial imputation or a smart way to inform the encoder about this pattern, and thus falls beyond the scope of this paper.
> On the other hand, ensemble or particle filter approaches typically rely on an established dynamics model (as discussed in line 82-88). While we have such a dynamics model available for the advection-diffusion dataset, applying ensemble or particle filtering would simply converge towards the true posterior distribution, which we have access to, and therefore would, in our opinion, not form a very useful comparison.
>
> > Though the posterior was evaluated in the manuscript, the dynamical model (F) wasn't. It is very often that you got a decent posterior but learned a "bad" F (in terms of forecasting).
>
> This is indeed a very interesting point. Unfortunately, we haven't had the time to evaluate the learned transition models, but we are hoping to do so during the discussion period.
>
> > Only the posterior mean was evaluated. It is unclear how good the posterior variance is.
>
> We do evaluate the posterior variance in terms of the CRPS metric. However, as the CRPS is a combined evaluation of the mean and variance, we now also include direct evaluations of the marginal standard deviations, for the advection-diffusion dataset where the true posterior distribution is available (see Table 1 in the provided PDF).
>
> > The variational posterior should be compared to the true posterior in an example where the latter is accessible analytically or via sampling.
>
> Thank you for this good suggestion. We will include this comparison in the appendix of the camera ready paper. To do this, we will perform the same evaluation as is done with the final posterior estimate (see above).
>
> > Only one synthetic example is not sufficient for evaluating the method thoroughly. It is unclear how it generalizes.
>
> We have added additional experiments on a real-world air quality monitoring dataset, showing that our approach generalizes well to more complex real world settings. For more details, please see the general response to all reviewers.
>
> ## Responses to Questions
>
> > Were the observation parameters $H$ and $\xi$ trainable or fixed at the true values?
> > Were the hyperparameters such as state noise and observation noise trainable or fixed at the true values?
>
> Thank you for pointing out that this crucial information is missing. In all experiments, the observation model parameters $\mathbf{H}$ and $\mathbf{R}$ ($\boldsymbol{\xi}$ is the random variable, not a parameter) are fixed. In particular, $\mathbf{H}$ is a selection matrix defined according to the training/validation/test masks used in each experiment, and the observation noise is defined as $\mathbf{R}=\sigma^2\mathbf{I}$. For the advection-diffusion experiments, $\sigma$ matches the value used to generate the data, while for the newly added air quality dataset it is set to $\sigma=0.01$. We will add this information to the description of the experimental setup.
>
> ## Conclusion
> We thank reviewer 9F96 again for taking the time to provide thoughtful feedback and for pointing out some missing information about the experimental setup. We hope that our responses and actions taken w.r.t. the evaluation of posterior variances and additional experiments on a real-world dataset are satisfactory. If there are any other changes you would like to see, we are more than happy to discuss them with you.

---

> > ### Comment · Reviewer_9F96 · 2023-08-11
> >
> > Thank the authors for the response.
> >
> > > We added a spatiotemporal ST-AR baseline (see general rebuttal) to address this limitation. Unfortunately, other more complex filter/smoother approaches are more difficult to apply in the setting we are considering. For example, deep Kalman filtering with KVAE [17] requires the definition of a suitable encoder mapping from high-dimensional observations to a latent space. In settings where the pattern of missing observations varies over time, this requires some initial imputation or a smart way to inform the encoder about this pattern, and thus falls beyond the scope of this paper. On the other hand, ensemble or particle filter approaches typically rely on an established dynamics model (as discussed in line 82-88). While we have such a dynamics model available for the advection-diffusion dataset, applying ensemble or particle filtering would simply converge towards the true posterior distribution, which we have access to, and therefore would, in our opinion, not form a very useful comparison.
> > - It could be the example to compare with the true posterior.
> > - Missing observations is not a must for every example, is it?
> > - There are variational methods learning the, e.g. [Frigola 2014](https://papers.nips.cc/paper_files/paper/2014/hash/139f0874f2ded2e41b0393c4ac5644f7-Abstract.html) and [Naesseth 2017](https://arxiv.org/abs/1705.11140). Dual Kalman filter could also learn the dynamical model.
> >
> > Fig 1. bottom. Three methods perfectly overlap for input data?

---

> > > ### Author Response · Authors · 2023-08-17
> > >
> > > >- It could be the example to compare with the true posterior.
> > >
> > > To do this, we have now implemented the Ensemble Kalman Smoother with an advection-diffusion transition model matching the data-generating process. We used $10^4$ ensemble members (the maximum feasible on our machine). Fixing the velocity and diffusion parameters to the true values, we obtain the following results:
> > >
> > > $MAE_{\mu}$ = 0.0512$\tiny{\pm 0.0019}$, $RMSE_{\mu}$ = 0.0654$\tiny{\pm 0.0028}$, $MAE_{\sigma}$ = 0.0041$\tiny{\pm 0.0000}$, $RMSE_{\sigma}$ = 0.0045$\tiny{\pm 0.0000}$, $CRPS$ = 0.1025$\tiny{\pm 0.0029}$
> > >
> > > As expected, the estimated posterior is very close to the true posterior.
> > >
> > > For a more appropriate comparison with our approach, we used a state augmentation approach to estimate the velocity and diffusion parameters jointly with the system states. In contrast to the ST-DGMRF approach, we consider initial and transition noise parameters to be fixed in order to avoid divergence of the EnKS. This yields the following results:
> > >
> > > $MAE_{\mu}$ = 0.1249$\tiny{\pm 0.1423}$, $RMSE_{\mu}$ = 0.1925$\tiny{\pm 0.2504}$, $MAE_{\sigma}$ = 0.0046$\tiny{\pm 0.0011}$, $RMSE_{\sigma}$ = 0.0061$\tiny{\pm 0.0031}$, $CRPS$ = 0.1624$\tiny{\pm 0.1226}$
> > >
> > > While $MAE_{\sigma}, RMSE_{\sigma}$ are lower than with our approach, $MAE_{\mu}, RMSE_{\mu}$ are slightly higher than with the ST-DGMRF model with $L_{\text{temporal}}=4$.
> > > Note that $MAE_{\sigma}$ and $RMSE_{\sigma}$ are expected to increase as well when considering the noise parameters unknown.
> > >
> > > Reducing the ensemble size by half, increases $MAE_{\mu}$ and $RMSE_{\mu}$ further to 0.1647$\tiny{\pm} 0.1274$ and 0.2244$\tiny{\pm 0.1765}$ respectively.
> > >
> > > >- Missing observations is not a must for every example, is it?
> > >
> > > You are right, missing observations are not a must. However, settings with partially observed system states are a focus of our paper and one of the main motivations for developing our approach (see lines 15-25), which is why we designed our experiments accordingly.
> > >
> > > >- There are variational methods learning the, e.g. [Frigola 2014](https://papers.nips.cc/paper_files/paper/2014/hash/139f0874f2ded2e41b0393c4ac5644f7-Abstract.html) and [Naesseth 2017](https://arxiv.org/abs/1705.11140). Dual Kalman filter could also learn the dynamical model.
> > >
> > > Thank you for sharing these very interesting papers with us. Using Gaussian processes in combination with variational inference, as done in Frigola 2014, is indeed a very promising approach. However, their application seems (so far) limited to small systems (e.g. one variable and its derivative). Applying such an approach to large systems with interacting components would require designing a suitable GP transition with both high-dimensional input and high-dimensional output. This seems beyond the scope of this paper.
> > >
> > > >Fig 1. bottom. Three methods perfectly overlap for input data?
> > >
> > > Yes, this is expected because the input data points $\mathbf{y}$ are given to all these methods at inference time. Unless the observation noise is set very high, the estimated posterior $p(\mathbf{x}\mid\mathbf{y})$ will thus be very close to the observed values at these points for all considered methods.
> > >
> > > If you have any further questions, we would be happy to answer them.

---

> > > > ### Author Response · Authors · 2023-08-18
> > > > **Evaluation of learned dynamics**
> > > >
> > > > > Though the posterior was evaluated in the manuscript, the dynamical model (F) wasn't. It is very often that you got a decent posterior but learned a "bad" F (in terms of forecasting).
> > > >
> > > > We have now compared the learned transitions for different ST-DGMRF models trained on the advection-diffusion dataset with the true data-generating process, which we have access to in this case. In particular, we extracted the stencil for an arbitrary point on the lattice (all points are equivalent) from the true transition matrix as well as from the learned ones. We then compared them qualitatively by visualizing the local graph structure and the corresponding weights of the stencil. Finally, we computed the Pearson correlation coefficient and the MAE between learned and ground truth weights.
> > > >
> > > > We find that for all ST-DGMRF variants the learned transition model is very close to the ground truth. The more temporal layers are used, the better the agreement. Moreover, even with only 2 temporal layers we learn an adequate approximation of the true transition, with Pearson correlation coefficients remaining above 0.87 (p << 0.01) and MAE remaining below 0.025 (ground truth weights range between -0.26 and 0.36), for both the advection-diffusion and the neural network transition models respectively. This is in good agreement with the corresponding posterior evaluation (Table 1 in the provided PDF).
> > > >
> > > > We are planning to incorporate these results in the main text of the paper.
> > > >
> > > > (also replied to k13T)

---

> > > > > ### Comment · Reviewer_9F96 · 2023-08-18
> > > > >
> > > > > Thank you for the responses. You have addressed my questions. I have raised the score to 6.

---

### Author Rebuttal · Authors · 2023-08-10

# General rebuttal to all reviewers

First of all, we would like to thank all the reviewers for their thoughtful and constructive feedback. We are delighted to hear that our paper is "clear" (LAkB), "easy to read" (Upf6) and "professionally written" (k13T), and that the reviewers found our proposed method to be "well explained" (Upf6), "well motivated" (Upf6), "principled and well derived" (k13T) and "quite a good idea" (Upf6).

Concerns were raised mainly about the experimental evaluation, which was limited to a single simulated dataset and to relatively simple (purely temporal and purely spatial) baseline models.
Here, we provide a short summary of the major actions we have taken to address these concerns.

1. We agree with the reviewers that it is important to show how our method generalizes to real world settings. Therefore, we performed additional experiments on real air quality sensor data.
2. We acknowledge the lack of a suitable spatiotemporal baseline in our evaluation. To address this limitation, we have included an AR process with unconstrained spatial noise in the transition model (as suggested by reviewer LAkB) as an additional baseline. We are happy to report that our method can improve on the performance of this spatiotemporal baseline.
3. Reviewers pointed out that while the true posterior is available for the simulated advection-diffusion dataset, we did not actually use it in the performance evaluation. This is a good point, and we have adjusted Table 2 accordingly.
4. Reviewer k13T correctly pointed out that the performance gain reported in the submitted paper is good but incremental. We are happy to report that we were able to improve the performance of all ST-DGMRF variants by a significant amount (see Table 1). An important ingredient was to replace the standard CG method by a regularized version [60], which allows for more stable posterior estimation, in particular for ST-DGMRFs with large number of temporal layers.

In the following, we provide detailed information on each of these points. Minor changes and clarifications are addressed in the responses to the individual reviewers.

## Additional experiments

We performed additional experiments on real air quality sensor data obtained from [58].
The dataset contains hourly PM2.5 measurements taken at 246 sensors distributed around Beijing. We considered a time period of T=400 hours in March 2015 and extracted relevant weather covariates from the ERA5 reanalysis [59]. The base graph is defined based on the Delaunay triangulation of sensor locations.

### Experimental setup

To define our test set, we randomly draw 10 time points $t_k$ and mask out all measurements within a spatial block (containing 50% of all sensors) for time steps $t_k, …, t_k+20$, mimicking partial network failures.

Since the transport of pollutants is a complex process involving many different temporal scales, we hypothesize that both increasing the Markov order $p$ and the number of temporal layers will improve the estimated posterior. To test this, and to validate that ST-DGMRF is indeed able to capture such higher order dependencies, we perform an ablation study where we vary $L_{\text{temporal}}$ from 1 to 4, for Markov order $p=1$ and $p=2$ respectively.

We consider two ST-DGMRF variants: one with simple "diffusion" layers, and one neural network variant taking edge features and weather covariates as inputs.

### Main results (see Table 2 and Figure 1)

- We find that ST-DGMRF estimates unobserved system states more accurately than all considered baseline models.
- As expected, increasing $p$ from 1 to 2 results in more accurate posterior estimates, for both variants.
- Similarly, increasing $L_{\text{temporal}}$ results in continuous improvement in posterior estimates, for both variants.

## Additional baseline models

The spatiotemporal AR model with unconstrained spatial noise (ST-AR) takes the form
$x_k = \alpha\cdot x_{k-1} + \epsilon_k$, where $x_0 \sim \mathcal{N}(\mu_0, \Sigma_0)$ and $\epsilon_k \sim \mathcal{N}(0, \mathbf{Q}^{-1})$.

We fix $\Sigma_0 = 10\cdot I$ to encode high uncertainty about the initial state $x_0$, and fit $\alpha, \mu_0$ and $\mathbf{Q}^{-1}$ to data using closed-form EM updates. We initialize the EM algorithm with $\alpha=1$, $\mu=\mathbf{0}$ and $\mathbf{Q}^{-1}=diag(\mathbf{q})$ where elements $\mathbf{q}_i \in [3, 4]$ are initialized randomly. After convergence of the EM-algorithm, the final state estimates are obtained with the Kalman smoother.

## Additional evaluations

For the advection-diffusion dataset, for which the true posterior distribution is available, we now evaluate the estimated posterior mean and std w.r.t their ground truth.
That means, we replaced the MAE and RMSE computed based on the system state (Table 2 in the submitted paper) by $MAE_{\mu}$, $RMSE_{\mu}$, $MAE_{\sigma}$ and $RMSE_{\sigma}$ (see Table 1).

## General improvements

We replaced the standard CG method with a regularized CG variant [60], resulting in more stable final posterior estimations.
This allowed us to improve the performance of ST-DGMRF variants with large numbers of temporal layers significantly (see Table 1). Note that in contrast to Table 2 in the submitted paper, showing results for $L_{\text{temporal}}=1$, we now show results for $L_{\text{temporal}}=2$ and $L_{\text{temporal}}=4$. As expected, the ST-DGMRF with advection-diffusion dynamics and $L_{\text{temporal}}=4$ performs best, as it has exactly the same form as the true transition model.

## References

[58] Y. Zheng et al. Forecasting fine-grained air quality based on big data. In Proceedings of the 21th SIGKDD conference on Knowledge Discovery and Data Mining. 2015.

[59] H. Hersbach et al. The ERA5 global reanalysis. Quarterly Journal of the Royal Meteorological Society. 2020

[60] Bai ZZ, Zhang SL. A regularized conjugate gradient method for symmetric positive definite system of linear equations. Journal of Computational Mathematics. 2002

---

### Decision · Program_Chairs · 2023-09-21

**Decision:**

Accept (poster)

**Comment:**

The four reviewers of this paper found the work well written, the methods principled, and the approach interesting. On the other side, initially, there were multiple concerns related to experimental validation (lack thereof), the somewhat restricted setting, and the black-box nature of the results. However, following the discussion most of the concerns were addressed in a satisfactory manner. The reviewers recommend accepting this work, with the only reviewer weakly recommending rejection having a low confidence score. During the rebuttal/discussion, the authors promised to add several things to their paper (incl. additional experiments). Go through the promised changes in detail and add them to the camera-ready version of your paper.